# Inhibition of GSK3α,β rescues cognitive phenotypes in a preclinical mouse model of CTNNB1 syndrome

Jonathan M Alexander [1], Leeanne Vazquez-Ramirez [1], Crystal Lin[1], Pantelis Antonoudiou[1], Jamie Maguire[1], Florence Wagner[2,3] & Michele H Jacob [1]✉

## Abstract

**CTNNB1 syndrome is a rare monogenetic disorder caused by *CTNNB1* de novo pathogenic heterozygous loss-of-function variants that result in cognitive and motor disabilities. Treatment is currently lacking; our study addresses this critical need. *CTNNB1* encodes β-catenin which is essential for normal brain function via its dual roles in cadherin-based synaptic adhesion complexes and canonical Wnt signal transduction. We have generated a *Ctnnb1* germline heterozygous mouse line that displays cognitive and motor deficits, resembling key features of CTNNB1 syndrome in humans. Compared with wild-type littermates, *Ctnnb1* heterozygous mice also exhibit decreases in brain β-catenin, β-catenin association with N-cadherin, Wnt target gene expression, and Na/K ATPases, key regulators of changes in ion gradients during high activity. Consistently, hippocampal neuron functional properties and excitability are altered. Most important, we identify a highly selective inhibitor of glycogen synthase kinase (GSK)3α,β that significantly normalizes the phenotypes to closely meet wild-type littermate levels. Our data provide new insights into brain molecular and functional changes, and the first evidence for an efficacious treatment with therapeutic potential for individuals with CTNNB1 syndrome.**

**Keywords** CTNNB1; β-Catenin; GSK3; Cognition; Motor
**Subject Category** Neuroscience

## Introduction

CTNNB1 syndrome is a neurodevelopmental disorder characterized by intellectual disability (ID), microcephaly, global developmental delay (motor, language), and motor disabilities (truncal hypotonia, distal hypertonia, peripheral spasticity, poor coordination, and muscle weakness). It is a monogenetic disorder caused by de novo pathogenic heterozygous loss-of-function variants in *CTNNB1*, resulting in haploinsufficiency (de Ligt et al, 2012; Dubruc et al, 2014; Kharbanda et al, 2017; Kuechler et al, 2015; Li

et al, 2017; Sudnawa et al, 2024; Tucci et al, 2014; Winczewska-Wiktor et al, 2016). The motor and cognitive disabilities severely impact the quality of life for these individuals. There are currently no treatment options beyond supportive care, with assistance for daily functioning often required.

CTNNB1 syndrome was first identified in 2012. To date, 500 children have been definitively genetically diagnosed, with males and females affected in equal numbers. Further, recent studies identify *CTNNB1* pathogenic variants as a lead genetic cause in individuals misdiagnosed with cerebral palsy (Deciphering Developmental Disorders, 2017; Jin et al, 2020; Kayumi et al, 2022; Srivastava et al, 2022; Takezawa et al, 2018). Thus, CTNNB1 syndrome prevalence estimates are increasing as more children with relevant symptoms undergo genetic testing, with current estimates of 1 in 50,000 births (Lopez-Rivera et al, 2020).

Human genetic studies identify *CTNNB1* as a high-confidence risk gene for ID and autism (https://gene.sfari.org/database/human-gene/CTNNB1) (Caracci et al, 2021; Cederquist et al, 2020; de Ligt et al, 2012; Deciphering Developmental Disorders, 2017; Dubruc et al, 2014; Hormozdiari et al, 2015; Iossifov et al, 2014; Krumm et al, 2014; Kuechler et al, 2015; O'Roak et al, 2012a; O'Roak et al, 2012b; Packer, 2018; Satterstrom et al, 2019; Tucci et al, 2014). The *CTNNB1* gene encodes the protein β-catenin (β-cat), which is essential for normal brain development and function, via its dual roles in cadherin-based synaptic adhesion complexes and as a transcription co-activator with LEF/TCF proteins in the canonical Wnt signal transduction pathway (Ataman et al, 2008; Brigidi and Bamji, 2011; Chen et al, 2006; Mills et al, 2014; Murase et al, 2002; Okuda et al, 2007; Park and Shen, 2012; Rosso and Inestrosa, 2013; Salinas, 2012; Tanaka et al, 2012; Yu and Malenka, 2003). Further, other ID- and autism-linked human gene mutations also cause reduced β-cat levels or function, including *CHD8* (Durak et al, 2016), *TCF7L2* (Dias et al, 2021), *Shank3* (Hassani Nia et al, 2020; Ioannidis et al, 2023; Qin et al, 2018), and *Arx* (Cho et al, 2017). The significant disease relevance of this pathway highlights the importance of gaining new insights into pathophysiological changes caused by reduced β-cat and identifying corrective treatments.

To address this critical unmet medical need, we have generated a *Ctnnb1* germline heterozygote (het) mouse designed to model human CTNNB1 syndrome caused by germline pathogenic variants resulting in haploinsufficiency. We show that the *Ctnnb1* het mouse

[1]Tufts University School of Biomedical Sciences, Department of Neuroscience, Boston, MA 02111, USA. [2]The Broad Institute of MIT and Harvard, Center for the Development of Therapeutics, Cambridge, MA 02142, USA. [3]Present address: Photys Therapeutics, Waltham, MA, USA. ✉E-mail: Michele.Jacob@Tufts.edu

exhibits cognitive and motor impairments resembling key features of human CTNNB1 syndrome. We provide new insights that have high translational significance for this global heterozygote disorder. We elucidate novel molecular and functional changes in the het mouse brain, compared with wild-type littermates. Most important, we identify a highly selective small-molecule inhibitor of glycogen synthase kinase (GSK)3α,β as an efficacious therapeutic treatment. Inhibiting this endogenous negative regulator of β-cat levels in vivo raises the insufficient β-cat levels in the het mouse brain and skeletal muscle, to closely meet, and not exceed, the normal baseline range of wild-type littermates, thereby correcting the root pathological change in this monogenetic disorder. We demonstrate significant improvement of the molecular, functional, motor and cognitive phenotypes. Further, the therapeutic treatment provides corrective outcomes when administered in symptomatic late adolescent/ young adult mice. Our findings of an efficacious treatment in the preclinical het mouse provide the first evidence for a potential therapeutic that may ameliorate the disabilities in individuals with CTNNB1 syndrome.

# Results

## A new mouse model of Ctnnb1 haploinsufficiency

We have developed a preclinical in vivo mouse model of human CTNNB1 syndrome by germline deletion of one allele of *Ctnnb1* that encodes the protein β-cat. Whole body homozygous deletion of Ctnnb1 is embryonic lethal, and conditional heterozygous or homozygous deletion targeting specific cell types does not model the human disease of CTNNB1 germline haploinsufficiency. We crossed mice carrying *Ctnnb1* with loxP-flanked exons 2–6 (*Ctnnb1* fl/fl) (Brault et al, 2001; Wickham et al, 2019) with Ella/E2a-Cre mice (JAX Labs) (hybrid mix of background strains C57B6/J; 129/ SvJ; FVB; Fig. 1A). Cre-mediated excision of the loxP-flanked DNA sequence causes loss of functional β-cat protein (Brault et al, 2001; Grigoryan et al, 2013; Joksimovic and Awatramani, 2014; Wickham et al, 2019). The E2a promoter directs Cre recombinase expression in nearly all embryonic tissues, including germline (Lakso et al, 1996). Offspring positive for expression of Cre recombinase were then crossed to wild-type mice and their offspring were genotyped to identify mice that carried germline deletion of one *Ctnnb1* allele in the absence of E2a-Cre expression. The germline *Ctnnb1*$^{+/-}$ (β-cat$^{+/-}$) mice are bred to wild-type mice, to generate β-cat$^{+/-}$ (β-cat het) and β-cat$^{+/+}$ mice. The latter serve as littermate controls in all experiments.

To verify the deletion of one *Ctnnb1* allele, we performed quantitative real-time PCR from hippocampal tissue (Fig. 1B) and show that β-cat het mice express ~50% levels of Ctnnb1 mRNA. To determine if this heterozygote gene expression results in lowered expression of the β-cat protein, we performed quantitative immunoblots of the frontal cortex, hippocampal and skeletal muscle tissues (Figs. 1C and EV1). Compared with wild-type littermates, β-cat global het mice display approximately 50% reductions in β-cat protein levels, resembling previous reports that β-cat protein levels closely mirror mRNA levels (Rudloff and Kemler, 2012). This mouse model exhibits *Ctnnb1* haploinsufficiency, the core molecular change predicted for individuals with CTNNB1 syndrome.

## β-cat het mice display impaired cognitive and motor capabilities

As ID is a characteristic feature of CTNNB1 syndrome in humans, we assessed associative spatial learning and memory acquisition in the β-cat het mice using contextual fear conditioning and probe trial assays (Fig. 2A). Compared to wild-type littermates, β-cat het mice exhibit significantly reduced freezing behavior over the training trials and the 24-h probe trial for short-term memory. Plots of the individual mouse performance values across all trials show variability, but no consistently lowest levels per individual throughout the assay, and no significant differences between male and female mice (Fig. EV2).

Next, we tested whether β-cat het mice display altered motor capabilities, another salient feature of human CTNNB1 syndrome. We performed grip-strength tests to assess forelimb strength and rotarod tests to evaluate motor learning and coordination. Compared to littermate controls, β-cat het mice demonstrate reduced forelimb strength on an average of three trials (Fig. 2B). β-cat het mice also show impaired motor capabilities on the rotarod assay (Fig. 2C), failing to achieve the same latency to fall over three successive trials, relative to wild-type littermates.

General locomotion was not significantly altered in β-cat het mice, as assessed by average velocity (distance traveled over time) in the open field (Fig. 2D), eliminating a potential confound in the behavior performance experiments. Thus, β-cat het mice display core behavioral features associated with human CTNNB1 syndrome, establishing it as a valuable preclinical in vivo model for identifying pathological changes caused by *Ctnnb1* haploinsufficiency and corrective therapeutic strategies.

## β-cat het mice show decreases in synaptic adhesion molecular interactions and expression levels of canonical Wnt targets

As β-cat functions in two pathways critical to proper brain function and cognition—canonical Wnt signal transduction and cadherin synaptic adhesion complexes—we assessed whether the reduced β-cat levels may alter these molecular pathways, using quantitative immunoblot and immunoprecipitation analyses on the hippocampus of β-cat het mice. First, we measured the total levels of two key proteins that interact with β-cat at the synapse, N-cadherin and α-N-catenin, that function in transsynaptic adhesion and linking the synaptic adhesion complex to the submembranous actin cytoskeleton. We observed no difference in the total levels of either protein in β-cat het mice compared to control littermates (Fig. 3A). We then examined the total levels of LEF1, a β-cat nuclear co-transcription factor in the canonical Wnt pathway. We found no difference in LEF1 levels in β-cat het mice compared to littermate controls (Fig. 3A).

Using co-immunoprecipitation, we tested whether the reduced β-cat levels of β-cat het mice affected interactions with N-cadherin, its direct binding partner at excitatory synapses. β-cat association with N-cadherin is reduced in β-cat het mice, relative to littermate controls (Fig. 3B). Further, the reduced β-cat levels led to decreases in the protein levels of two canonical Wnt target genes, the Wnt antagonist Dickkopf WNT Signaling Pathway Inhibitor 1 (DKK1) and Engrailed Homeobox 2 (EN2), both known to affect cognitive function (Brielmaier et al, 2012; Marzo et al, 2016; Provenzano

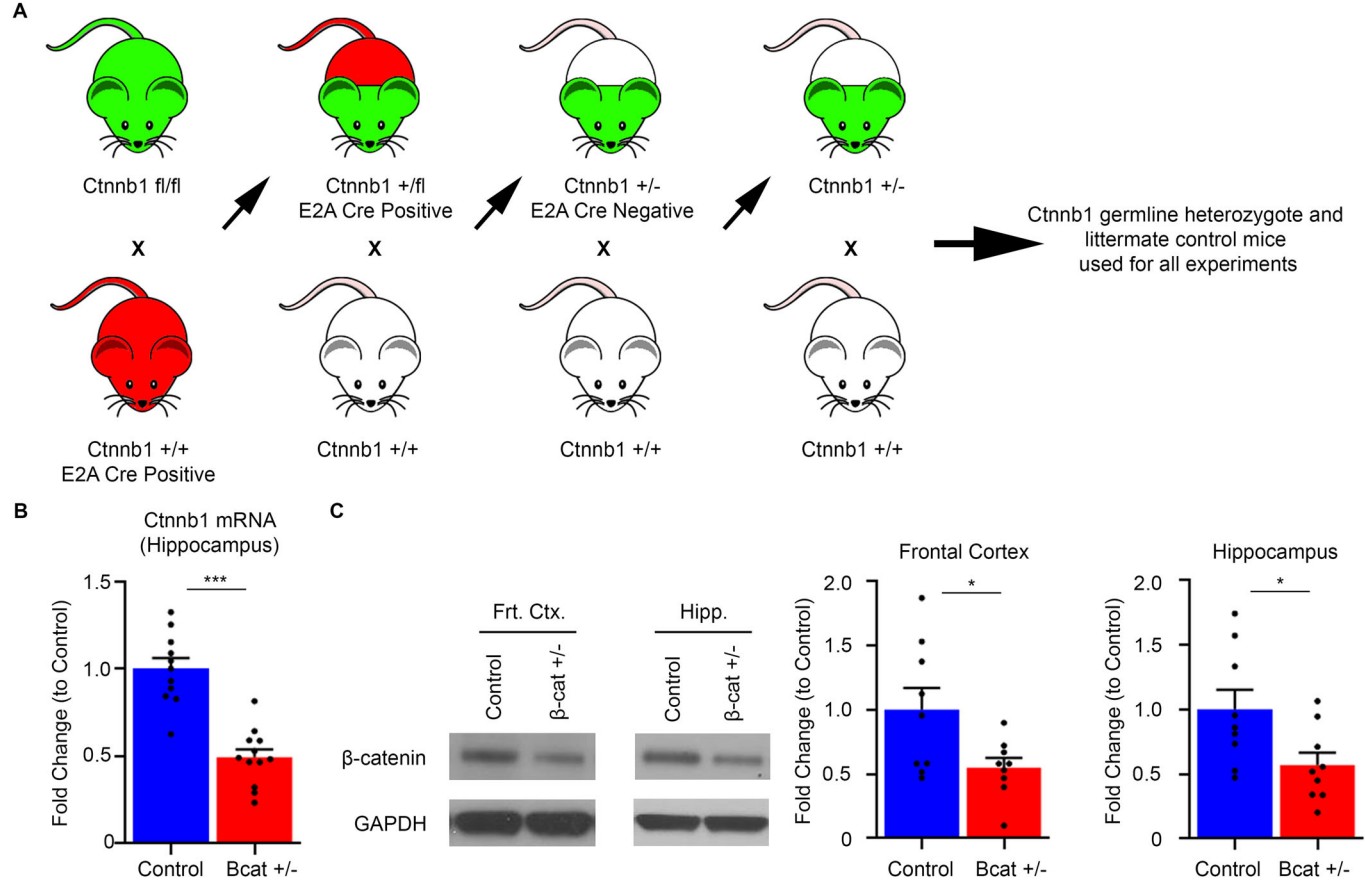

**Figure 1. A novel mouse model for CTNNB1 syndrome.**

(A) Schematic representation of the generation of the β-cat[+/−] mouse model. *Ctnnb1[fl/fl]* mice (green color) were crossed with E2A Cre+ mice (red color) resulting in germline deletion of one allele of the *Ctnnb1* gene. Germline heterozygous mice that carry the deletion in the absence of Cre recombinase are crossed with wild-type mice (no color) to generate β-cat[+/−] (β-cat het) and β-cat[+/+] as littermate controls. (B) Quantitative reverse-transcriptase PCR shows an approximate 50% reduction in *Ctnnb1* mRNA hippocampal levels ($n = 11$ Control mice, 12 β-cat het mice, Student's *t* test, ***$P < 0.001$). (C) Immunoblots and quantification show an approximate 50% reduction in β-catenin protein levels in the frontal cortex and hippocampus of β-cat het mice, compared to wild-type littermates. (Frontal cortex, $n = 9$ per genotype, Student's *t* test, *$P = 0.028$, Hippocampus, $n = 9$ per genotype, Student's *t* test, *$P = 0.027$.) All values are reported as mean of biological replicates $+/−$ s.e.m. from two independent experiments. Source data are available online for this figure.

et al, 2014) (Fig. 3C). Compared to wild-type littermates, the β-cat het mouse exhibits hippocampal molecular changes in synaptic adhesion and canonical Wnt signaling pathways that impact brain function.

## β-cat het mice exhibit altered functional properties and excitability in hippocampal neurons

To begin to gain insights into functional pathophysiological changes in the brain caused by *Ctnnb1* haploinsufficiency, we examined the electrophysiological properties and excitability of hippocampal dentate gyrus granule cells (DGGCs) in the β-cat het mice, compared to control littermates, by patch clamp analysis of acute hippocampal slices. We found that the normally hyperpolarized resting membrane potential of DGGCs is significantly more depolarized in β-cat hets, compared to controls (Fig. 4A). β-cat het DGGCs also exhibit impaired excitability with altered input–output curves, showing reduced spike frequency (max firing rate) at higher current injections (Fig. 4B). Impaired excitability is further evident

from analysis of the action potential waveforms showing reduced afterhyperpolarization (Fig. 4C) and reduced rheobase (Fig. 4D) in β-cat het mice, compared to wild-type littermates, suggesting altered ion channel function and action potential properties. Taken together, the functional changes suggest reduced threshold for excitability and potential depolarization block.

## β-cat het mice display altered levels of Na/K ATPases known to modulate neuronal functional properties

The family of Na/K ATPase transporters, key regulators of $Na^+$ $K^+$ ion gradients in excitable cells, is known to impact the functional properties that we find altered in the β-cat het mouse hippocampus, including the resting membrane potential, action potential waveform, afterhyperpolarization and excitability. While other channels such as HCN also modulate these functional properties, we focused on Na/K ATPases based on Na/K ATPase α3 association with the N-cadherin, β-cat synaptic adhesion complex in cultured hippocampal neurons (Tanaka et al, 2012) and similar changes, down

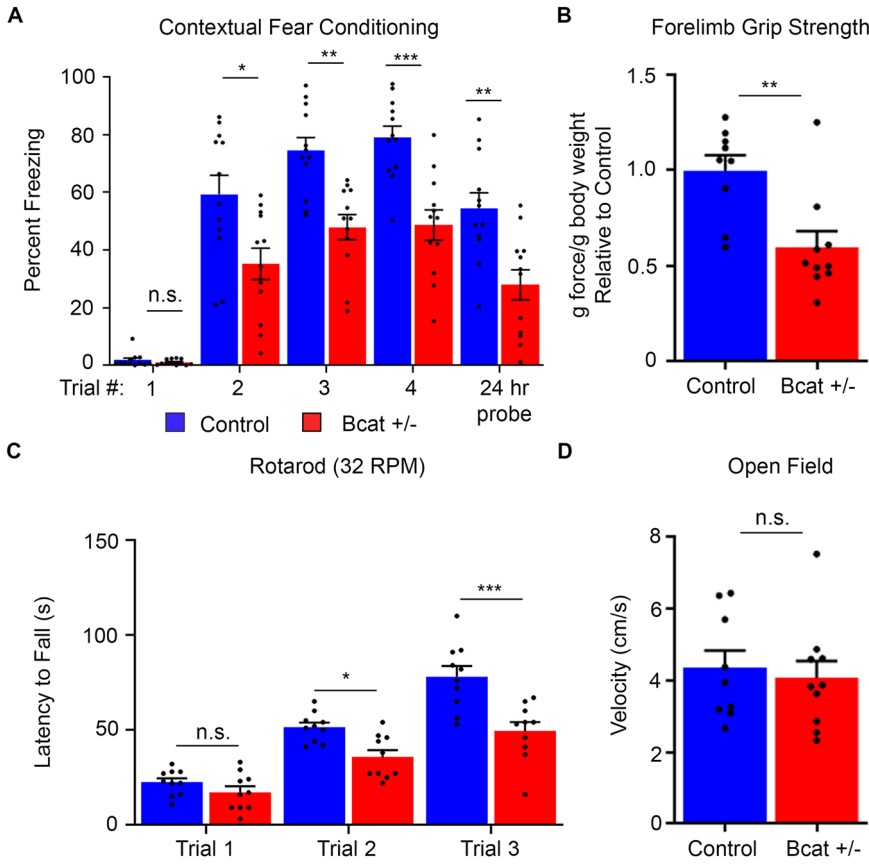

**Figure 2. β-cat het mice exhibit core behavioral phenotypes associated with CTNNB1 syndrome.**

Compared with wild-type littermates, β-cat het mice display (**A**) cognitive impairments based on their reduced freezing behavior over the training period and the 24 h probe test in the contextual fear conditioning task for associative spatial learning ($n = 12$ per genotype, repeated-measures ANOVA $F[1, 22] = 33.80$, $P < 0.001$; Trial 1: $P = 0.874$, Trial 2: *$P = 0.050$, Trial 3: **$P = 0.001$, Trial 4: ***$P < 0.001$, 24 h Probe: **$P = 0.009$, Bonferroni multiple $t$ test correction), (**B**) reduced forelimb grip strength in the average of three trials ($n = 9$ controls, 10 β-cat het, Student's $t$ test **$P = 0.003$), (**C**) deficits in motor learning and coordination, based on their reduced latency to fall over the training trials in the rotarod assay ($n = 10$ per genotype, repeated-measures ANOVA, $F[1,18] = 7.109$, $P = 0.016$; Trial 1: $P = 0.975$, Trial 2: *$P = 0.0439$, Trial 3: ***$P < 0.001$, Bonferroni multiple $t$ test correction), and (**D**) comparable locomotor activity (velocity of movement) in the open field task ($n = 9$ controls, 10 β-cat het, Student's $t$ test $P = 0.693$). All values are reported as the mean of biological replicates $+/-$ s.e.m. from two independent experiments. Source data are available online for this figure.

and up, between Na/K ATPase α2 and β-cat levels (via siRNA knockdown to 50% levels and overexpression) in C2C12 muscle cell lines (Zhao et al, 2019; Zhao et al, 2014). Three isoforms are expressed in the nervous system: α1, expressed in neurons and glia, the major regulator of $Na^+$ $K^+$ gradients at the resting state; α2, predominantly expressed in astrocytes in the adult brain, and α3, expressed in neurons, with α2 and α3 critical for $Na^+$ $K^+$ gradient changes during high activity (McGrail et al, 1991; Moseley et al, 2003; Radzyukevich et al, 2013; Sweadner, 1992). We measured the catalytic α subunit levels, as they are a key determining factor for the transporter expression and activity.

We show in the mouse brain in vivo, that β-cat heterozygosity results in decreases in Na/K ATPase α2 and α3 levels, compared to wild-type littermates, based on quantitative immunoblot analysis of the hippocampus and frontal cortex (Fig. 5). These molecular changes are consistent with the electrophysiological changes (Fig. 4). In contrast, there is no significant change in Na/K ATPase α1 subunit levels (Fig. 5), the isoform that maintains the resting ion gradients. The reductions in Na/K ATPases α2 and α3 in the β-cat

het brain suggest novel molecular targets for modulating activity-dependent function changes.

## Small-molecule inhibition of GSK3 significantly normalizes molecular changes in the β-cat het brain

An endogenous negative regulator of β-cat levels, glycogen synthase kinase (GSK)3 is a critical component of the β-cat destruction complex, it phosphorylates β-cat, marking it for ubiquitination and proteasomal degradation (Gordon and Nusse, 2006). GSK3 has two isoforms, α and β, both have been implicated in synaptic plasticity and cognitive function (Kondratiuk et al, 2017; McCamphill et al, 2020; Shahab et al, 2014). We tested whether inhibiting the endogenous GSK3 isoforms, either together or separately, might increase the reduced β-cat levels and thereby improve phenotypes of the β-cat het mice. We employed the latest generation of small-molecule GSK3 inhibitors, BRD0320 (both α and β paralogs), BRD0705 (α selective), and BRD3731 (β selective). BRD0705 and BRD3731 show 8-14-fold selectivity for GSK3 α and β, respectively

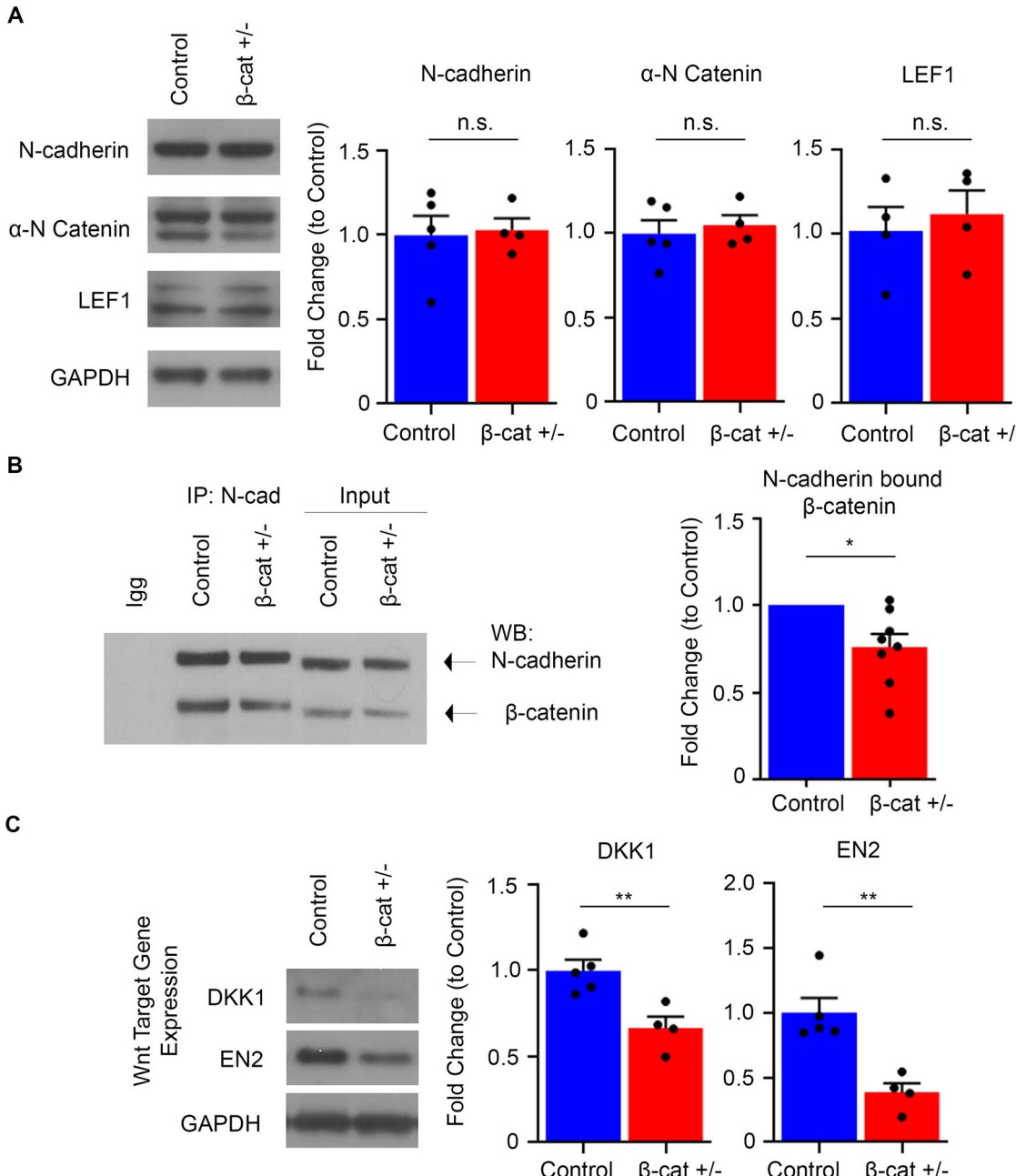

**Figure 3. β-cat het mice display decreased N-cadherin-β-cat synaptic complex interactions and expression levels of Wnt targets.**

Representative immunoblots and quantification show (**A**) no changes in the total protein levels of β-cat binding partners, N-cadherin, α-N-catenin, and Lef1 in the β-cat het hippocampus ($n = 5$ controls, 4 β-cat het; N-cadherin: Student's $t$ test, $P = 0.825$, α-N-catenin: Student's $t$ test, $P = 0.674$, LEF1: $n = 4$ each genotype, Student's $t$ test, $P = 0.552$), relative to wild-type littermates. GAPDH, as a loading control. (**B**) Co-immunoprecipitation of β-cat with N-cadherin show reduced interactions in the β-cat het hippocampus ($n = 8$ per genotype; one-sample $t$ test, $*P = 0.021$). (**C**) Protein expression levels of two canonical Wnt target genes, DKK1 and EN2, are reduced in the β-cat het hippocampus ($n = 5$ controls, 4 β-cat het; DKK1: Student's $t$ test, $P = 0.008$, EN2: Student's $t$ test, $P = 0.003$). All values are reported as mean of biological replicates +/− s.e.m. from two independent experiments. Source data are available online for this figure.

(Wagner et al, 2018; Wagner et al, 2016). We chose these particular GSK3 inhibitors based on their advantageous characteristics of exquisite specificity for the target kinase against the greater kinome, brain permeability (brain/plasma ratio of 0.16), and favorable pharmacokinetic (PK)/pharmacodynamic (PD) profiles in in vivo mouse studies (Bernard-Gauthier et al, 2019; McCamphill et al,

2020; Wagner et al, 2018; Wagner et al, 2016). We performed intraperitoneal injections (30 mg/kg) once daily for five days in β-cat het mice at 6–10 weeks of age (late adolescence/ young adult stage) and assessed the phenotypes starting 1 h after the last injection, with the cognitive probe test assay for memory acquisition extending to 24 h post-treatment (Figs. 6A and 8A).

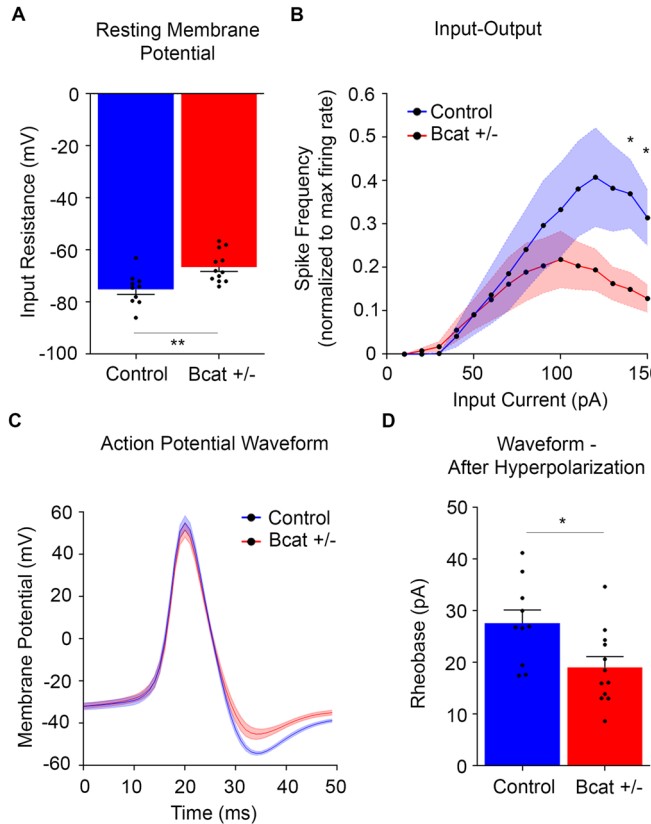

**Figure 4.  β-cat het mice show altered functional properties in hippocampal principal neurons.**

Compared to wild-type littermates, β-cat het mice show impaired excitability in patch clamp analysis of dentate gyrus granule cells (DGGCs) in acute hippocampal slices, based on (A) significantly more depolarized resting membrane potential ($n = 5$ controls, 6 β-cat het mice, 2 cells each mouse; Student's $t$ test, **$P = 0.003$), (B) altered input–output curves with reduced spike frequency (reduced max firing rate) at higher current injections ($n = 5$ controls, 6 β-cat het mice, 2 cells each mouse, Fisher's LSD test, 140pA: *$P = 0.022$; 150 pA: *$P = 0.023$), (C) altered action potential waveform with reduced afterhyperpolarization and (D) reduced rheobase ($n = 5$ controls, 6 β-cat het mice, 2 cells each mouse, *$P = 0.016$, Student's $t$ test). All values are reported as mean of biological replicates $+/-$ s.e.m. from two independent experiments. Source data are available online for this figure.

Previous studies show that at this dose, each compound is present in the brain at a concentration above their in vitro half-maximal inhibitory concentration ($IC_{50}$) for at least 4 h after dosing (McCamphill et al, 2020). The GSK3 inhibitor-treated hets were compared to vehicle-treated het and vehicle-treated wild-type littermates.

Immunoblot analysis shows significant increases in β-cat levels in both the hippocampus and frontal cortex, as well as skeletal muscle (Fig. EV1), of dual-paralog inhibitor-treated β-cat het mice, compared to vehicle-treated β-cat het mice (Fig. 6B). Importantly, the increased β-cat levels closely meet, and are not higher or significantly different from the normal baseline range of vehicle-treated wild-type littermates (Fig. 6B). Further, the reduced Na/K ATPase α2 and α3 levels are also significantly normalized by the dual-paralog inhibitor in the β-cat het hippocampus and frontal cortex, compared with vehicle-treated β-cat het and wild-type

littermate levels (Fig. 6B). In comparison, Na/K ATPase α1 levels are not altered by the dual inhibitor treatment (Fig. 6B).

As further correction of the molecular changes, GSK3 dual inhibitor-treated β-cat het mice exhibited an increased association of β-cat with N-cadherin, not significantly different from that of littermate controls, in immunoprecipitation assays from the hippocampus (Fig. 6C). Further, β-cat het mice treated with the GSK3 dual inhibitor displayed increased protein expression levels of the Wnt targets DKK1 and EN2, not significantly different from littermate control levels (Fig. 6D).

To determine whether inhibition of both GSK3 paralogs was required for these corrective outcomes, we tested the efficacy of α and β selective GSK3 inhibitors, using the same treatment protocol (Fig. 6A). Both were less effective at correcting the molecular changes. Treating with the α selective GSK3 inhibitor did not significantly increase β-cat levels, relative to vehicle-treated β-cat het mice (Fig. EV3A). The β selective GSK3 inhibitor significantly increased β-cat levels, relative to vehicle-treated β-cat het mice, to more closely resemble wild-type littermate levels (Fig. EV3B). However, Na/K ATPase α2 and α3 levels were not significantly increased by the β selective inhibitor relative to the vehicle-treated β-cat hets (Fig. EV3B).

Taken together, the results show that small-molecule inhibition of both GSK3α,β isoforms in vivo corrects the molecular changes, including β-cat, Na/K ATPase α2, α3, β-cat-N-cadherin co-immunoprecipitation, and Wnt target expression levels in the brain of β-cat haploinsufficient mice.

## GSK3α,β dual inhibitor improves excitability changes in the β-cat het hippocampus

Based on correction of the molecular changes, we assessed the effect of the dual-paralog inhibitor on excitability changes in the β-cat het hippocampal DGGC neurons by patch clamp analysis of acute hippocampal slices starting at 1 h after the last injection of the 5-day GSK3 inhibitor treatment regimen (Fig. 6A). Compared to vehicle-treated het DGGCs, the resting membrane potential was not altered (Fig. 7A). Input–output curves show significantly improved excitability, with increased spike frequency/ max firing rate at higher current injections in the GSK3 dual inhibitor-treated hets (Fig. 7B). Further, the GSK3 dual inhibitor increased the action potential waveform after hyperpolarization (Fig. 7C) and rheobase (Fig. 7D). The improved excitability of the GSK3α,β inhibitor-treated het mice is not significantly different from the excitability of wild-type littermates, based on comparisons of spike frequency (max firing rate) at higher current injections and relative levels of rheobase currents necessary to elicit an action potential (Fig. EV4).

## GSK3α,β inhibitor significantly improves the cognitive and motor capabilities of β-cat het mice

Based on the corrected molecular and functional changes in the β-cat het mouse brain treated with the dual-paralog inhibitor, we tested for improvements in the cognitive and motor phenotypes. We used the same treatment paradigm as that used for the molecular and excitability rescues, followed at the end of the 5-day treatment by assaying contextual fear conditioning starting at an hour after the last injection and probe trials 24 h later (Fig. 6A). The dual inhibitor (BRD0320) treated β-cat het mice displayed

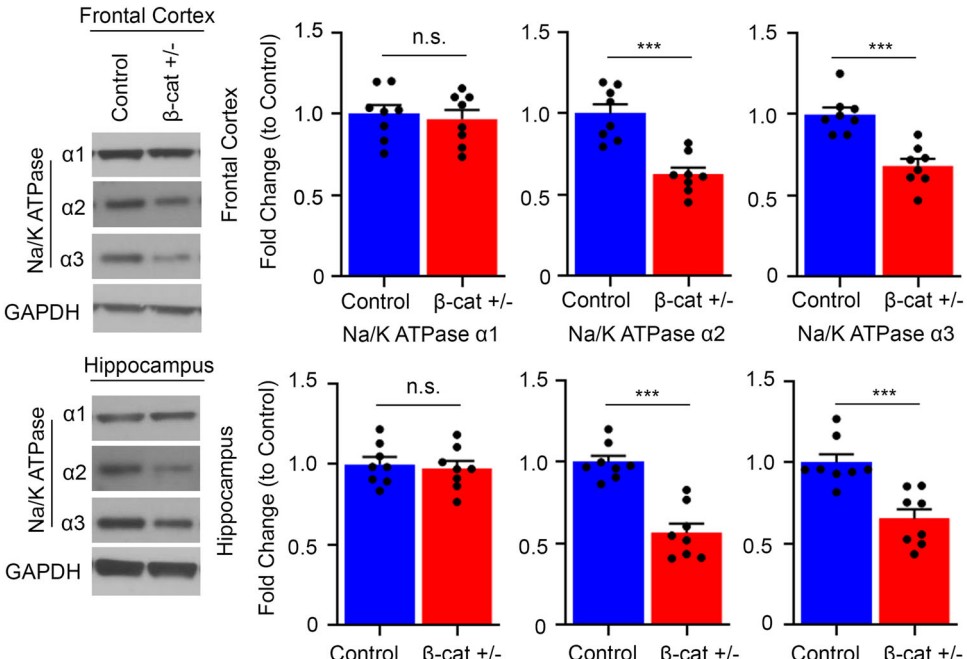

**Figure 5.** β-cat het mice exhibit reduced levels of Na/K ATPases, key regulators of Na$^+$ K$^+$ ion gradient changes that impact neuronal excitability.

Representative immunoblots and quantification show Na/K ATPases α2 and α3 catalytic subunit levels are reduced, whereas Na/K ATPase α1 catalytic subunit levels are not altered in the β-cat het hippocampus and frontal cortex, compared to wild-type littermates ($n = 8$ per genotype, α1 hippocampus: Student's $t$ test, $P = 0.695$; α1 frontal cortex: Student's $t$ test, $P = 0.694$; α2, α3: hippocampus and frontal cortex: Student's $t$ test, ***$P < 0.001$). GAPDH, as a loading control. All values are reported as mean of biological replicates $+/-$ s.e.m. from two independent experiments. Source data are available online for this figure.

significant improvements in their performance over the four training trials, compared to vehicle-treated β-cat het mice, and no significant differences relative to wild-type littermates (Fig. 8A). The cognitive improvement was retained 24 h after training and post-treatment, as shown in the probe test for short-term memory.

Further, the GSK3α,β inhibitor-treated β-cat het mice exhibited improved motor learning and coordination in the rotarod assay. They showed significantly longer latency to fall compared with vehicle-treated het littermates, and latencies not significantly different from vehicle-treated wild-type littermates (Fig. 8B). Forelimb grip strength was also significantly improved in the GSK3α,β inhibitor-treated β-cat hets, relative to the vehicle-treated hets (Fig. 8C). Moreover, the inhibitor treatment did not cause anxiety or a change in general locomotion or velocity of movement in the β-cat het mouse, as assessed in the elevated-plus maze task (Fig. 8D,E).

For comparison, we also treated wild-type littermates with the dual inhibitor, using the same treatment paradigm as for the β-cat hets (Fig. 6A). They exhibited no significant changes in either cognitive performance in the contextual fear conditioning and 24-h probe trial tasks or in β-cat protein levels in the hippocampus (Fig. EV5).

To assess whether separately inhibiting each GSK3 isoform might be sufficient to significantly improve the cognitive phenotype of the β-cat het mice, we used the same treatment paradigm with either BRD0705 (α selective) or BRD3731 (β selective) injected IP for 5 days at 30 mg/kg, followed 1 h after the last injection with cognitive testing in the contextual fear conditioning and 24-h probe trial tasks. Similar to the limited efficacy of either the α- or

β-selective GSK3 inhibitor to correct the molecular changes compared with the dual-paralog inhibitor, neither selective inhibitor was as effective in improving the cognitive capabilities of the β-cat het mice. While the α-selective inhibitor-treated β-cat hets exhibited no significant differences in freezing relative to vehicle-treated β-cat het littermates, the β-selective inhibitor-treated β-cat het mice displayed a small significant increase in cognitive capabilities, based on greater freezing during later training trials, relative to vehicle-treated het littermates (Trial 4 Percent Freezing; Fig. EV6). Further, neither the α- nor β-selective inhibitor led to improved performance in the 24-h probe trial, with no significant differences from vehicle-treated β-cat het littermates, while remaining significantly decreased compared with vehicle-treated wild-type littermates (Fig. 8A). Thus, selective inhibition of either GSK3 α or β isoform alone appears to be insufficient to remedy the cognitive phenotype, at least using the same treatment paradigm that is effective with the dual-paralog inhibitor. Our findings identify GSK3α,β dual-paralog inhibition as an efficacious treatment for significantly remedying molecular, functional, cognitive and motor phenotypes caused by β-cat heterozygosity.

## Discussion

Our data provide two major findings on CTNNB1 syndrome: (1) we elucidate novel molecular and functional changes in the brain that advance our understanding of the pathophysiology of this disorder, and (2) we provide the first evidence for a therapeutic treatment that normalizes phenotypes relevant to this unmet

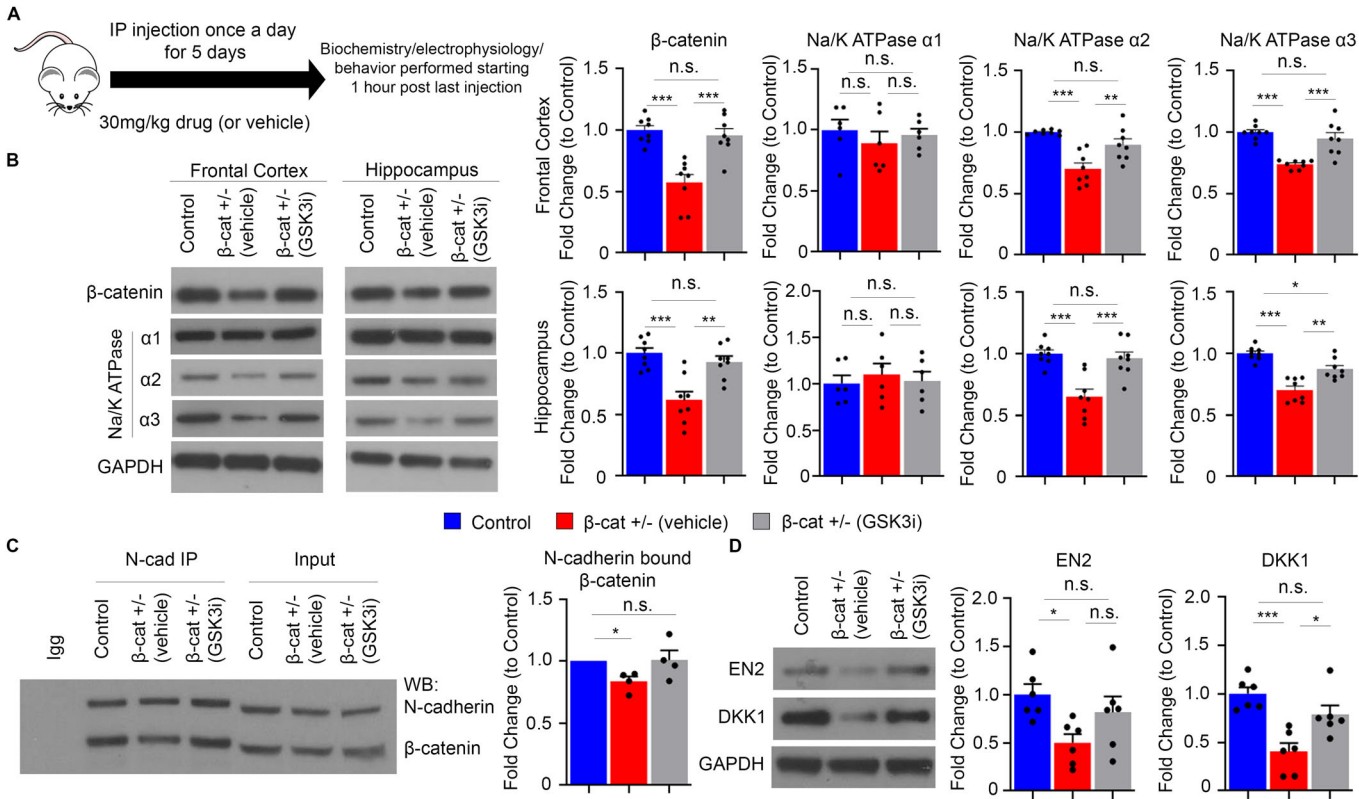

**Figure 6. Dual inhibition of GSK3α,β paralogs significantly normalizes molecular changes in β-cat het mice.**

(A) Schematic of the GSK3 α/β inhibitor treatment paradigm in 6–10 week-old β-cat het mice. (B) Immunoblots and quantification show that the dual-paralog inhibitor (GSK3i) normalizes β-cat levels in both the frontal cortex and hippocampus of β-cat het mice, compared to vehicle-treated β-cat het and vehicle-treated wild-type littermates, with GAPDH, as a loading control ($n = 8$ per genotype/condition, frontal cortex: one-way ANOVA $F[2,21] = 18.46$; $P < 0.001$, Tukey's multiple $t$ test correction Ctl to β-cat het veh ***$P < 0.001$, β-cat het veh to β-cat het GSK3i ***$P<0.001$, Ctl to β-cat het GSK3i $P = 0.849$; Hippocampus: one-way ANOVA $F[2,21] = 13.92$; $P < 0.001$, Tukey's multiple $t$ test correction Ctl to β-cat het veh ***$P < 0.001$, β-cat het veh to β-cat het GSK3i **$P = 0.002$, Ctl to β-cat het GSK3i $P = 0.639$). The dual inhibitor also significantly normalizes Na/K ATPase α2 and α3 levels in the β-cat het mice, whereas Na/K ATPase α1 levels are not altered by the dual inhibitor (frontal cortex: α1—$n = 6$ each genotype/condition, one-way ANOVA $F[2,15] = 0.4811$; $P = 0.627$, Tukey's multiple $t$ test correction Ctl to β-cat het veh $P = 0.606$, β-cat het veh to β-cat het GSK3i $P = 0.816$, Ctl to β-cat het GSK3i $P = 0.932$; α2 —$n = 8$ each genotype/condition, one-way ANOVA $F[2,21] = 15.21$; $P < 0.001$, Tukey's multiple $t$ test correction Ctl to β-cat het veh ***$P < 0.001$, β-cat het veh to β-cat het GSK3i **$P = 0.005$, Ctl to β-cat het GSK3i $P = 0.183$; α3—$n = 8$ each genotype/condition, one-way ANOVA $F[2,21] = 19.66$, $P < 0.001$, Tukey's multiple $t$ test correction Ctl to β-cat het veh ***$P < 0.001$, β-cat het veh to β-cat het GSK3i ***$P < 0.001$, Ctl to β-cat het GSK3i $P = 0.470$. Hippocampus: α1—$n = 6$ each genotype/condition, one-way ANOVA $F[2,15] = 0.2187$; $P = 0.806$, Tukey's multiple $t$ test correction Ctl to β-cat het veh p = 0.801, β-cat het veh to β-cat het GSK3i $P = 0.888$, Ctl to β-cat het GSK3i $P = 0.984$; α2—$n = 8$ each genotype/condition, one-way ANOVA $F[2,21] = 14.58$; $P < 0.001$, Tukey's multiple $t$ test correction Ctl to β-cat het veh ***$P < 0.001$, β-cat het veh to β-cat het GSK3i ***$P < 0.001$, Ctl to β-cat het GSK3i $P = 0.865$; α3—$n = 8$ each genotype/condition, one-way ANOVA $F[2,21] = 27.77$; $P<0.001$ Tukey's multiple $t$ test correction Ctl to β-cat het veh ***$P < 0.001$, β-cat het veh to β-cat het GSK3i **$P = 0.001$, Ctl to β-cat het GSK3i *$P = 0.012$). (C) Immunoprecipitation with N-cadherin shows that treating β-cat het mice with the GSK3 dual inhibitor increases N-cadherin bound β-cat to levels resembling that of wild-type littermates ($n = 4$ each genotype/condition, one-sample $t$ test β-cat het veh to Ctl *$P = 0.028$, β-cat het GSK3i to Ctl $P = 0.912$). (D) GSK3 dual inhibitor-treated β-cat het mice also show increases in protein expression levels of Wnt targets DKK1 and EN2, not significantly different from control levels ($n = 6$ each genotype/condition, DKK1— one-way ANOVA $F[2,15] = 4.010$; $P = 0.40$ Tukey's multiple $t$ test correction Ctl to β-cat het veh *$P = 0.034$, β-cat het veh to β-cat het GSK3i $P = 0.215$, Ctl to β-cat het GSK3i $P = 0.567$; EN2- one-way ANOVA $F[2,15] = 12.46$; $P < 0.001$ Tukey's multiple $t$ test correction Ctl to β-cat het veh ***$P < 0.001$, β-cat het veh to β-cat het GSK3i *$P = 0.017$, Ctl to β-cat het GSK3i $P = 0.210$). All values are reported as mean of biological replicates $+/-$ s.e.m. from two independent experiments. Source data are available online for this figure.

medical need. We have generated a new mouse line of *Ctnnb1* germline heterozygosity that displays cognitive and motor impairments that resemble key features of human CTNNB1 syndrome, establishing it as a useful preclinical in vivo model. We identify the previously unrecognized molecular change of decreases in Na/K ATPases α2 and α3 in the β-cat het mouse brain. We propose that these decreases likely play a significant role in the pathophysiology, based on the critical roles of these transporters in regulating ion gradient changes that modulate excitability and synaptic transmission, consistent with the functional changes that we have identified

in the β-cat het neurons (reduced max firing rate, rheobase, and afterhyperpolarization). Further, we show changes in β-cat networks critical for normal brain function, consisting of decreases in β-cat co-immunoprecipitation with N-cadherin transsynaptic adhesion molecule and protein expression levels of two canonical Wnt targets known to impact cognition. Most importantly, we provide the first evidence of the potential for a corrective treatment. We define β-cat and its endogenous negative regulator, GSK3, as critical molecular targets for efficacious therapeutic intervention. We show that in vivo inhibition of GSK3α,β paralogs brings the

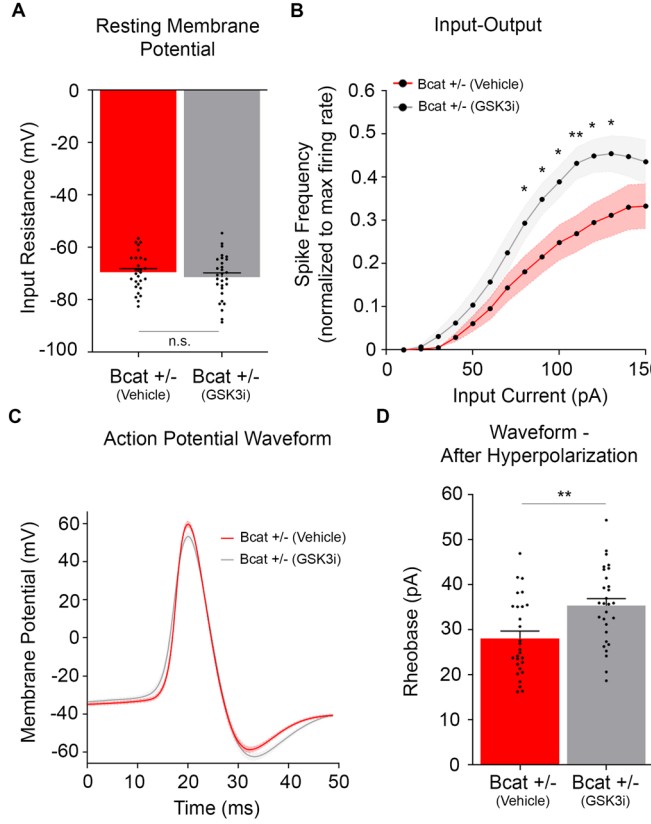

**Figure 7. Dual inhibition of GSK3α,β paralogs improves the altered functional properties and excitability of β-cat het mice.**

Patch clamp analysis of dentate gyrus granule cells (DGGCs) in acute hippocampal slices from GSK3i-treated β-cat het mice, compared to vehicle-treated β-cat het mice. (A) The resting membrane potential was not significantly altered by the dual inhibitor ($n = 10$–12 mice per genotype/condition, 2–3 cells each mouse; Student's $t$ test, $P = 0.372$). (B) Input–output curves show improved spike frequency (max firing rate) at higher current injections in GSK3i-treated β-cat het mice ($n = 10$–12 mice per genotype/condition, 2–3 cells each mouse; Fisher's LSD test, 80 pA: *$P = 0.040$; 90 pA: *$P = 0.015$; 100 pA: *$P = 0.014$; 110 pA: **$P = 0.006$; 120 pA: *$P = 0.013$; 130 pA: *$P = 0.030$). The dual inhibitor also (C) increased the action potential waveform afterhyperpolarization and (D) rheobase, compared to vehicle-treated β-cat het mice ($n = 10$–12 mice per genotype/condition, 2–3 cells each mouse; Student's $t$ test, **$P = 0.002$). All values are reported as mean of biological replicates $+/-$ s.e.m. from two independent experiments. Source data are available online for this figure.

reduced β-cat levels of β-cat het mice to within, and not above, the normal baseline range of wild-type littermates. The GSK3α,β inhibitor also significantly improves Na/K ATPase α2 and α3 levels, β-cat-N-cadherin and Wnt pathway changes, and excitability changes in the brain. Moreover, learning and memory capabilities of the GSK3α,β treated β-cat het mice are statistically similar to that of wild-type littermates and significantly improved relative to vehicle-treated β-cat het littermates. Our data identify a new highly selective small-molecule GSK3α,β dual-paralog inhibitor as an in vivo efficacious treatment for correcting *Ctnnb1*/β-cat haploin-sufficiency phenotypes. Albeit dual inhibition of GSK3α,β paralogs has been hampered as a treatment strategy in clinical trials in disorders with normal β-cat levels due to β-cat stabilization and the potential for aberrant cell proliferation, increasing β-cat levels is the

goal for CTNNB1 syndrome where β-cat levels are reduced by pathogenic heterozygous loss-of-function variants. Moreover, the therapeutic treatment was effective when administered in symptomatic mice at 6–10 weeks old (late adolescence, young adult stages), the age required for reliable testing of complex cognitive behaviors in mice. These findings demonstrate that treatment initiated after the onset of symptoms provides significant corrective outcomes. The lack of treatment options for individuals with this severe disorder highlights the importance of our findings.

Precedence shows that other GSK3α,β inhibitors have advanced to Phase 2 trials in adults and Phase 2/3 trials in children and adolescents for other conditions (Rizzieri et al, 2016; Horrigan et al, 2020; Clinical trial NCT05004129). Further, support for the feasibility of long-term GSK3 inhibition at appropriate doses stems from individuals with psychiatric disorders being treated for decades with lithium salts, which include GSK3 inhibition as one of its targets. Based on its high selectivity for GSK3 against the human kinome, blood brain barrier permeability and physico-chemical properties, we believe that our small-molecule GSK3 inhibitor is best for developing an efficacious and safe treatment for CTNNB1 syndrome patients.

Our novel finding of decreases in Na/K ATPases α2 and α3 in the β-cat het mouse brain suggests a key pathophysiological role by altering neuronal functional properties. These decreases in the β-cat het mouse brain in vivo extend previous reports that Na/K ATPase α3 associates with the N-cadherin- β-cat synaptic adhesion complex in cultured hippocampal neurons (Tanaka et al, 2012) and of similar changes between Na/K ATPase α2 with β-cat levels, both up- and down- (β-cat overexpression and siRNA knockdown to 50% levels) in the C2C12 muscle cell line (Zhao et al, 2019; Zhao et al, 2014). Further, the extent of the decreases in the β-cat het brain Na/K ATPases α2 and α3 (60–75% of wild-type littermate levels) closely resemble the muscle α2 decreases—(65–75% of control levels) that are sufficient to alter functional properties (Clausen and Everts, 1991; Radzyukevich et al, 2013; Sopjani et al, 2010; Zhao et al, 2019; Zhao et al, 2014). Na/K ATPase regulation of the $K^+$ and $Na^+$ ion gradients in cells is essential for normal synaptic transmission and excitability of neurons and skeletal muscle. The heteromeric enzyme is an integral membrane protein composed of catalytic α, regulatory β, and an auxiliary FXYD subunit (Blanco et al, 1998). The α subunits are encoded by different genes and exhibit tissue-specific expression; α1 is ubiquitous; α2 is restricted to skeletal muscle, smooth muscle, heart, and brain, where it is predominantly in astrocytes; and α3 is expressed in excitatory and inhibitory neurons (Bottger et al, 2011; McGrail et al, 1991; Moseley et al, 2003; Smith et al, 2021). Na/K ATPase α1 is the major housekeeping form and maintains basal intracellular $Na^+$ levels in the resting state, while α2 and α3 modulate large transient increases in intracellular $Na^+$ during high activity. In particular, α3 is critical in neurons for recovery of $K^+$ and $Na^+$ gradients following excitation. Aberrant ion gradients caused by Na/K ATPase malfunction or reduced levels (haploinsufficiency) alter neuronal synaptic transmission and excitability. The changes resemble those we detected in the β-cat het mouse hippocampus, including depolarization of the resting membrane potential, altered input resistance, altered action potential waveform, reduced afterhyperpolarization, and decreases in the maximum firing frequency to injected current, suggesting a reduced threshold for excitability and potential depolarization block

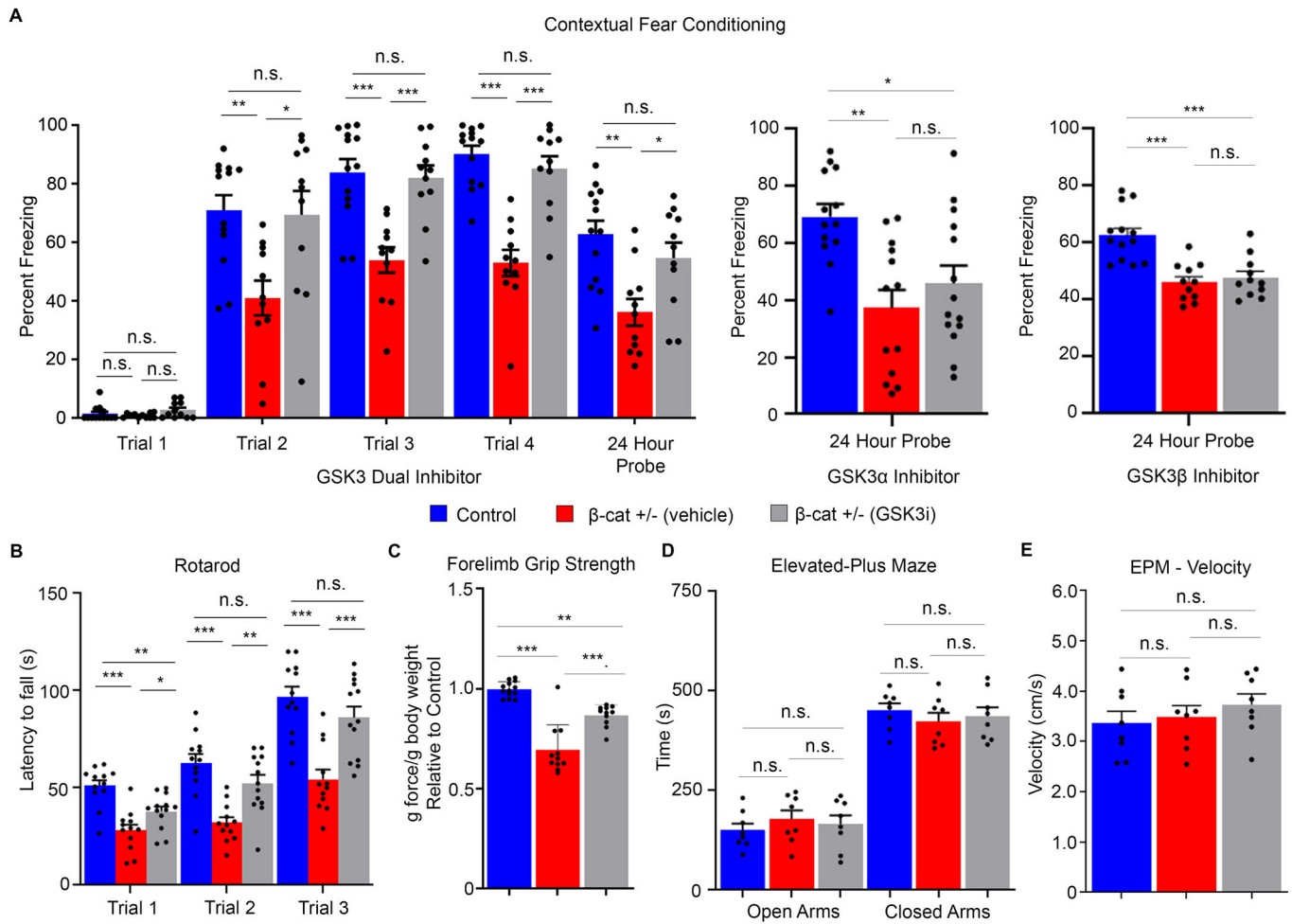

**Figure 8.  Dual-paralog Inhibitor of GSK3α,β significantly improves the cognitive and motor phenotypes of β-cat het mice.**

(A) GSK3α,β dual-paralog inhibition (BRD0320) significantly improves learning and memory in the β-cat het mice, based on increased freezing in the training trials and 24-h probe test, relative to vehicle-treated β-cat het mice. Their performance is not significantly different from that of wild-type littermates. ($n = 13$ vehicle-treated Control littermates, 11 β-cat het vehicle, 11 β-cat het dual inhibitor; repeated-measures ANOVA $F[2,32] = 23.19$, $P < 0.001$; Tukey's multiple comparison $t$ test Trial 1: Control vs. β-cat het vehicle, $P = 0.724$; Control vs. β-cat het dual inhibitor, $P = 0.513$, β-cat het vehicle vs. β-cat het dual inhibitor, $P = 0.142$; Trial 2: Control vs. β-cat het vehicle, **$P = 0.002$; Control vs. β-cat het dual inhibitor, $P = 0.985$, β-cat het vehicle vs. β-cat het dual inhibitor, *$P = 0.030$; Trial 3: Control vs. β-cat het vehicle, ***$P < 0.001$; Control vs. β-cat het dual inhibitor, $P = 0.954$, β-cat het vehicle vs. β-cat het dual inhibitor, ***$P < 0.001$; Trial 4: Control vs. β-cat het vehicle, ***$P < 0.001$; Control vs. β-cat het dual inhibitor, $P = 0.613$, β-cat het vehicle vs. β-cat het dual inhibitor, ***$P < 0.001$; 24 h Probe: Control vs. β-cat het vehicle, **$P = 0.002$; Control vs. β-cat het dual inhibitor, $P = 0.501$, β-cat het vehicle vs. β-cat het dual inhibitor, *$P = 0.038$). In comparison, the learning and memory deficits are not corrected by the α selective GSK3 inhibitor (BRD0705) or the β selective GSK3 inhibitor (BRD3731) as shown in the 24-h probe trial test after training. (α selective, BRD0705: $n = 13$ Control and vehicle-treated β-cat het, 14 α inhibitor-treated β-cat het mice; one-way ANOVA $F[2,37]$ $P = 0.001$; Bonferroni multiple $t$ test correction, Control vs. β-cat het vehicle, **$P = 0.002$; Control vs. β-cat het α inhibitor, *$P = 0.018$, β-cat het vehicle vs. β-cat het α inhibitor, $P > 0.9999$; β selective, BRD3731: $n = 13$ Controls, 11 β-cat het vehicle, 11 β-cat het β inhibitor; one-way ANOVA $F[2,32] = 73.32$ $P < 0.001$; Bonferroni multiple $t$ test correction, Control vs. β-cat het vehicle, $P < 0.001$; Control vs. β-cat het β inhibitor, ***$P < 0.001$, β-cat het vehicle vs. β-cat het β inhibitor, $P = 0.776$). (B) β-cat het mice treated with the GSK3 dual inhibitor displayed significantly improved performance in the rotarod test, compared to vehicle-treated β-cat het mice, especially at later trials ($n = 12$ Controls, 12 β-cat het vehicle, 13 β-cat het dual inhibitor; repeated-measures ANOVA $F[2,34] = 20.55$ $P < 0.001$; Tukey's multiple correction $t$ test Trial 1: Control vs. β-cat het vehicle, ***$P < 0.001$; Control vs. β-cat het dual inhibitor, **$P = 0.008$, β-cat het vehicle vs. β-cat het dual inhibitor, *$P = 0.050$; Trial 2: Control vs. β-cat het vehicle, ***$P < 0.001$; Control vs. β-cat het dual inhibitor, $P = 0.239$, β-cat het vehicle vs. β-cat het dual inhibitor, **$P = 0.002$; Trial 3: Control vs. β-cat het vehicle, ***$P < 0.001$; Control vs. β-cat het dual inhibitor, $P = 0.379$, β-cat het vehicle vs. β-cat het dual inhibitor, ***$P < 0.001$). (C) Dual inhibition of GSK3α,β also significantly improved forelimb grip strength in β-cat het mice ($n = 13$ Controls, 11 β-cat het vehicle, 11 β-cat het vehicle dual inhibitor, One-way ANOVA F[2,32] = 42.81, $P < 0.001$; Tukey's multiple correction $t$ test Control vs. β-cat het vehicle, ***$P < 0.001$; Control vs. β-cat het dual inhibitor, **$P = 0.001$, β-cat het vehicle vs. β-cat het dual inhibitor, ***$P < 0.001$). (D) β-cat het mice, both vehicle- and dual inhibitor-treated, showed no anxiety ($n = 8$ each genotype/condition, one-way ANOVA $F[2,21] = 0.499$ $P = 0.614$) or (E) impaired locomotion (velocity and distance) ($n = 8$ each genotype/condition, one-way ANOVA $F[2,21] = 0.678$ $P = 0.519$) as measured in the elevated-plus maze task. All values are reported as mean of biological replicates $+/-$ s.e.m. from two independent experiments. Source data are available online for this figure.

(Chakraborty et al, 2017; Vaillend et al, 2002). Moreover, Na/K ATPase α2 and α3 heterozygote mice and humans exhibit several phenotypes that resemble CTNNB1 syndrome. Individuals with heterozygous loss-of-function ATP1A3/ Na/K ATPase α3 variants display developmental delay, postnatal microcephaly, spasticity, axial hypotonia, and learning deficits (Holm et al, 2016; Holm and Lykke-Hartmann, 2016; Smith et al, 2021; Sweney et al, 2015). Individuals with ATP1A2/ Na/K ATPase α2 mutations often exhibit cognitive impairments, ataxia, sensory disturbances, and seizures (Bottger et al, 2016; Isaksen and Lykke-Hartmann, 2016). Mice heterozygous for α3 exhibit spatial learning and memory deficits, motor deficits, and increased locomotor activity (Holm and Lykke-Hartmann, 2016; Kirshenbaum et al, 2013; Moseley et al, 2007; Ng et al, 2021). Mice heterozygous for α2 display impaired spatial learning, increased anxiety-related behavior, reduced locomotor activity and hypercontractile skeletal muscle (He et al, 2001; Ikeda et al, 2003; Moseley et al, 2007). In contrast to α2 and α3 heterozygotes, Na/K ATPase α1 heterozygous mice show no change in learning. Our findings identify Na/K ATPases α2 and α3 as new molecular targets in CTNNB1 syndrome. Our future studies will assess additional molecular changes such as HCN channels that may contribute to excitability changes in the β-cat het brain.

Our data identify the highly selective GSK3α,β dual-paralog inhibitor (BRD0320) as an efficacious therapeutic treatment by showing that it significantly normalizes the molecular, excitability, cognitive and motor changes in the in vivo mouse model of Ctnnb1 germline heterozygosity. GSK3α,β paralogs are derived from different genes, and have both redundant and unique substrates and functions (Cormier et al, 2023; Liang and Chuang, 2006; Soutar et al, 2010). Both paralogs have been implicated in synaptic plasticity and cognitive function. While they do not modulate basal synaptic transmission, the GSK3 paralogs differentially modulate the balance between long-term potentiation (LTP) and long-term depression (LTD), electrophysiological correlates of synaptic plasticity required for learning and memory. GSK3α, but not GSK3β, is required for NMDAR-LTD in the wild-type mouse hippocampus (Draffin et al, 2021; Ebrahim Amini et al, 2022), as partial decreases (40–50%) in GSK3α reduced NMDAR-LTD and thereby enhanced LTP (Ebrahim Amini et al, 2022). Inhibition of GSK3β facilitates induction of LTP and blocks induction of LTD (Hooper et al, 2007; Peineau et al, 2009; Peineau et al, 2007), whereas overexpression of GSK3β inhibits LTP (Hooper et al, 2007). The rate of spatial memory acquisition is enhanced by dual inhibition of GSK3α,β paralogs (inhibitor CT99021) in wild-type mice (Lee et al, 2021). Selective inhibition of GSK3α corrected learning and memory deficits in the mouse model of Fragile X syndrome, the most prevalent inherited monogenetic cause of ID and autism (McCamphill et al, 2020). Selectively inhibiting GSK3α alone did not improve the cognitive phenotype of β-cat het mice. Further, it did not alter β-cat levels in the β-cat het brain, as previously seen for GSK3α inhibition in other in vivo mouse brain and in vitro human cell studies (McCamphill et al, 2020; Wagner et al, 2018). Selectively inhibiting GSK3β alone was partially effective at improving β-cat levels and cognitive performance in the β-cat hets. In comparison, GSK3α,β dual inhibition normalizes the insufficient β-cat levels and associated phenotypes in the Ctnnb1 germline heterozygous mouse. Similarly, increasing β-cat levels required silencing 3 of the 4 GSK3 alleles in mouse embryonic stem cells (Doble et al, 2007). Our data suggest that the combined inhibition of both GSK3 paralogs is required for the corrective outcomes.

We propose that targeting the root pathophysiological change of reduced β-cat levels and its associated functions mediates the significant beneficial outcomes. However, we cannot rule out a contribution of other β-cat independent GSK3α,β substrates. We are not able to decipher whether the corrective outcomes achieved with the dual-paralog inhibitor stem from a greater level of inhibition than that achieved with inhibiting the single isoforms alone, or from the combined effects of substrates targeted by GSK3α and GSK3β inhibition. The selective substrates of the GSK3 paralogs are as yet poorly defined (Sutherland, 2011).

We show significant beneficial outcomes with treatment of β-cat het mice at late adolescence/ young adult ages when they are symptomatic. Individuals with CTNNB1 syndrome display phenotypes starting in the first months of life, including global developmental delay (motor and language). Additional studies are needed to test whether the age of treatment, earlier or later ages, may affect outcomes, and to evaluate the safety of chronic treatment with the highly selective GSK3α,β inhibitor. As a first step, we show no significant molecular or cognitive changes in wild-type mice with the same dual inhibitor treatment paradigm that provided corrective outcomes in the β-cat het mice.

In summary, our findings provide the first evidence of the potential for efficacious treatment of the severe disorder of CTNNB1 syndrome. Our data identify β-cat and Na/K ATPases α2 and α3 as disease relevant molecular targets. We identify the highly selective GSK3α,β inhibitor as an in vivo therapeutic capable of significantly normalizing the pathophysiological changes and cognitive disabilities caused by β-cat germline haploinsufficiency in a preclinical mouse model that displays key features of human CTNNB1 syndrome.

# Methods

**Reagents and tools table**

| Reagent/resource | Reference or source | Identifier or catalog number |
|---|---|---|
| **Experimental models** | | |
| B6.129-Ctnnb1tm2Kem/ KnwJ | JAX Mice | Stock# 004152 |
| B6.FVB-Tg(EIIa-cre) C5379Lmgd/J | JAX Mice | Stock# 003724 |
| **Antibodies** | | |
| Mouse anti-β-catenin | ThermoFisher Scientific | Cat# 13-8400 |
| Mouse anti-N-cadherin | ThermoFisher Scientific | Cat# 33-3900 |
| Rabbit anti-LEF1 | Proteintech | Cat# 14972-1-AP |
| Rabbit anti-α-N-catenin | Cell Signaling | Cat# 2131 |
| Rabbit anti-DKK1 | Santa Cruz Biotechnology | Cat# sc-25516 |
| Goat anti-EN2 | Abcam | Cat# ab45867 |
| Rabbit anti-ATP1A1 | Proteintech | Cat# 14418-1-AP |
| Rabbit anti-ATP1A2 | Sigma Millipore | Cat# 07-674 |

| Reagent/resource | Reference or source | Identifier or catalog number |
| --- | --- | --- |
| Mouse anti-ATP1A3 | ThermoFisher Scientific | Cat# MA3-915 |
| Mouse anti-GAPDH | Sigma Millipore | Cat# 33-3900 |
| Mouse Igg1 Kappa | ThermoFisher Scientific | Cat# 14-4714-85 |
| **Oligonucleotides and other sequence-based reagents** | | |
| *Ctnnb1* – Fwd/Rev | PrimerBank | ID: 6671684a1 |
| GAPDH – Fwd/Rev | PrimerBank | ID: 6679937a1 |
| **Chemicals, enzymes, and other reagents** | | |
| GSK3α,β inhibitor | The Broad Institute | BRD-K87550320-001-07-3 |
| GSK3α selective inhibitor | The Broad Institute | BRD-K00760705-001-06-9 |
| GSK3β selective inhibitor | The Broad Institute | BRD-K21263731-001-06-7 |
| DMSO | Sigma | Cat# D2650 |
| PEG-400 | Rigaku | Cat# 1008415 |
| Normal saline | G Biosciences | Cat# 786-560 |
| Pierce BCA Protein Assay | ThermoFisher | Cat# 23227 |
| Halt Protease/ Phosphatase Inhibitor Cocktail | ThermoFisher | Cat# 78440 |
| NuPage 4-12% Bis/Tris Gels | ThermoFisher | Cat# NP0321 |
| iBlot2 Nitrocellulose Transfer Stacks | ThermoFisher | Cat# IB23001 |
| Autoradiography Film – UltraCruz | Santa Cruz Biotechnology | Cat# sc-201697 |
| Bovine Serum Albumin | Sigma | Cat# A7906 |
| Protein A/G agarose beads | Santa Cruz Biotechnology | Cat# sc-2003 |
| SYBR™ Green PCR Master Mix | ThermoFisher | Cat# 4309155 |
| NaCl | Sigma | Cat# S7653 |
| NaHCO$_3$ | Sigma | Cat# S5761 |
| NaH$_2$PO$_4$ | Sigma | Cat# 71507 |
| KCl | Sigma | Cat# P5405 |
| CaCl$_2$ | Sigma | Cat# C5670 |
| MgCl$_2$ | Sigma | Cat# M4880 |
| Dextrose | Sigma | Cat# G7021 |
| Kynurenic acid | Sigma | Cat# K3375 |
| HEPES | Sigma | Cat# H4034 |
| EGTA | Sigma | Cat# 03777 |
| Mg-ATP | Sigma | Cat# A9187 |
| Na-GTP | Sigma | Cat# 51120 |
| **Software** | | |
| Fiji ImageJ | https://downloads.imagej.net/fiji/latest/fiji-win64.zip | |
| GraphPad Prism V9.2 | https://www.graphpad.com/ | |
| Actimetrics Freeze Frame | https://actimetrics.com/products/freezeframe/ | |
| Noldus Ethovision XT | https://www.noldus.com/ethovision-xt | |

| Reagent/resource | Reference or source | Identifier or catalog number |
| --- | --- | --- |
| Conductor | Maze Engineers - Conduct Science | |
| MxPro | Agilent | |
| PowerLab | ADInstruments | |
| Python | https://www.python.org/ | |
| **Other** | | |
| Stratagene MX3000P QPCR System | Agilent | |
| VT1000S vibratome | Leica | |
| Axopatch 200B | Axon Instruments | |
| XCell SureLock™ Mini-Cell | ThermoFisher | |
| iBlot 2 Transfer System | ThermoFisher | |
| Mini-Medical Film Processor | AFP Manufacturing | |
| Rotarod | Maze Engineers - Conduct Science | |
| Habitest Fear Conditioning Chamber | Coulbourn Instruments | |
| Grip Strength Meter | Columbus Instruments | |

## Methods and protocols

### Animals

β-cat$^{fl/fl}$ (JAX Stock #004152) (Brault et al, 2001) and E2A-cre (JAX Stock #003724) (Lakso et al, 1996) mice were bred (see Fig. 1A) to produce germline deletion of one copy of the *CTNNB1* gene. Upon recombination, heterozygous mice were bred to controls, and cre-negative heterozygotes were selected for the establishment of the line. For all experiments, 6–10 week-old mice of both sexes were used with littermate *CTNNB1$^{+/+}$* mice used as controls. Mice are housed on a 12-h light/dark cycle.

### Behavior

Mice were housed on a reverse 12-h light/dark cycle and handled 5 min daily for a week before behavioral testing. Contextual fear conditioning was performed as previously described (Trouche et al, 2013) with the following modification: Intertrial intervals were spaced at 1 h for each subject to assess the acute effect of the GSK inhibitors on learning and memory, with the probe trial at 24 h after the last training trial. For each mouse, freezing scores were calculated by averaging freezing during minutes 2 and 3 of each trial. Freezing behavior was measured using a digital camera connected to a computer with Actimetrics Freeze Frame software. The bout length was 1 s and the threshold for freezing behavior was determined by an experimenter blind to experimental conditions, such as genotypes and drug- versus vehicle-treatments. Rotarod (Mohn et al, 2014), grip strength (Munier et al, 2022), elevated-plus maze (Holmes et al, 2003; Mohn et al, 2014) (Mohn), and open field tests (Wickham et al, 2019) were all performed as previously described.

### GSK inhibitors

BRD-K87550320-001-07-3 (BRD0320, both α and β), BRD-K00760705-001-06-9 (BRD0705, α selective), and BRD-K21263731-001-06-7

(BRD3731, β selective) were provided by the Broad Institute of MIT and Harvard. Compounds were dissolved in DMSO to 30 mg/ml. For intraperitoneal administration, stock compounds were further diluted to a formulation of 10% DMSO, 45% PEG-400, 45% saline (vehicle) and mice injected with 10 μl per gram of body weight (30 mg/kg). For comparisons, littermate control mice were injected with vehicle only. Mice were injected at 6–10 weeks old.

### Electrophysiology

Adult (8 week-old) β-cat het and littermate control mice were anesthetized with isoflurane and decapitated, and the brain was rapidly removed and placed immediately in ice-cold, oxygenated normal artificial cerebrospinal fluid [nACSF; containing (in mM) 126 NaCl, 26 NaHCO$_3$, 1.25 NaH$_2$PO$_4$, 2.5 KCl, 2 CaCl$_2$, 2 MgCl$_2$, and 10 dextrose (300–310 mosM)] containing 3 mM kynurenic acid and bubbled with 95% O$_2$–5% CO$_2$. Coronal hippocampal slices (350 μm thick) were prepared using a Leica VT1000S vibratome. Slices were allowed to recover in oxygenated nACSF for at least 1 h prior to transferring them to a recording chamber maintained at 33 °C (in-line heater; Warner Instruments) and perfused at a high flow rate (~4 ml/min) throughout the experiment. Input–output curves were generated as previously described by our laboratory (Lee and Maguire, 2013; MacKenzie and Maguire, 2015). The number of action potentials generated in response to a series of 500-ms current injections from 20 to 150 pA in 20-pA steps was measured in dentate gyrus granule cells (DGGCs) and CA1 pyramidal cells in the current-clamp configuration using an intracellular recording solution containing (in mM) 130 K-gluconate, 10 KCl, 4 NaCl, 10 HEPES, 0.1 EGTA, 2 Mg-ATP, and 0.3 Na-GTP (pH = 7.25, 280–290 mosM). Rheobase was calculated as the current to elicit the first action potential during a current ramp injection (50 pA injection over 10 s) and the action potential waveforms were compared using phase plane analysis where the first derivative of the voltage is plotted against the voltage as previously described (MacKenzie and Maguire, 2015). To estimate the current required to reach action potential threshold (iAP), a linear regression was fit to the first 4 non-identical data points. Input resistance was calculated using Ohm's law in response to negative current injections. Series resistance and whole-cell capacitance were continually monitored and compensated throughout the course of the experiment. Recordings were eliminated from data analysis if series resistance increased by >20%. For all electrophysiology experiments, data acquisition was carried out using an Axopatch 200B (Axon Instruments) and PowerLab hardware and software (AD Instruments). Data analysis was performed using analysis scripts developed in-house using Python.

### Biochemistry

qPCR was performed as previously described (Alexander et al, 2020). Primers (MGH PrimerBank) are as follows: Ctnnb1— Fwd: ATGGAGCCGGACAGAAAAGC, Rev: CTTGCCACTCAGGGAAGGA (PrimerBank ID: 6671684a1) and Gapdh—Fwd: AGGTCGGT GTGAACGGATTTG, Rev: TGTAGACCATGTAGTTGAGGTCA (PrimerBank ID: 6679937a1).

Immunoblots were performed as previously described (Mohn et al, 2014). The antibodies used are as follows: β-catenin (Thermo Fisher Scientific Cat# 13-8400, RRID:AB_2533039), GAPDH (Millipore Cat# MAB374, RRID:AB_2107445), N-cadherin (Thermo Fisher Scientific Cat# 33-3900, RRID:AB_2313779), LEF1 (Proteintech Cat# 14972-1-AP, RRID:AB_2265677), α-N-catenin (Cell Signaling Technology Cat# 2131, RRID:AB_2087922), DKK1 (Santa Cruz Biotechnology Cat# sc-25516, RRID:AB_2091346), EN2 (Abcam Cat# ab45867, RRID:AB_732160), ATP1A1 (Proteintech Cat# 14418-1-AP, RRID:AB_2227873), ATP1A2 (Millipore Cat# 07-674, RRID:AB_390164), APT1A3 (Thermo Fisher Scientific Cat# MA3-915, RRID:AB_2274447). Immunoblots were quantified using Fiji ImageJ using a standardized area of intensity measure and background subtracted. GAPDH was used as a loading control.

Immunoprecipitations of N-cadherin were done as follows: Protein A/G agarose beads (Santa Cruz Cat# sc-2003) were blocked with 1% BSA in RIPA buffer for an hour at 4 °C. Lysates were precleared with blocked beads for an hour at 4 °C. 1 mg of pre-cleared lysate was incubated with 2 μg N-cadherin antibody (see above) or Mouse Igg1 Kappa control (Thermo Fisher Scientific Cat# 14-4714-85, RRID:AB_470112) and blocked beads overnight at 4 °C. Beads were washed three times for 10 min and immunoprecipitate was eluted from the beads for immunoblotting with both N-cadherin and β-catenin antibodies (see above). Quantification of immunoprecipitated β-catenin was standardized the amount of N-cadherin immunoprecipitated with the control value for each blot set to a value of 1 for comparison to the β-cat het immunoprecipitation (either untreated (Fig. 3) or vehicle/GSK3i-treated (Fig. 6)). Because only one set of immunoprecipitates could be run per blot, a one-sample $t$ test was employed comparing the β-cat het immunoprecipitates to the control value of 1 in order to assess significance.

### Statistics

The sample size is based on similar experiments by us and our collaborators (Mohn et al, 2014; Wickham et al, 2019), the literature, and calculated to ensure power > 0.8, Alpha=0.05, Effect size=1.1. Biological replicates were randomized into experimental groups making sure that approximately equal number of sex/genotype/treatment were sampled from each litter and experimental animals were pooled from several litters to circumvent any home cage effects. Experimenters were blinded to genotype/treatment. Statistical analysis was carried out using Graphpad Prism (ver 9.2). The normality of the data was assessed using the Shapiro–Wilk test for normality. Outlier data was determined via the ROUT method, and no outliers are reported in this study. For single comparisons, Student's $t$ test were used. For multiple groups ANOVA tests were employed and post-hoc comparisons were done using a $t$ test corrected for multiple comparisons. All values are reported as a mean of biological replicates $+/-$ s.e.m. and the number of replicates and statistical analyses for all experiments are described in the figure legends.

### Study approval

All procedures involving animals were approved by the Tufts University Institutional Animal Care and Use Committee in accordance with National Institutes of Health guidelines.

## For more information

Please see CTNNB1 Connect and Cure: https://www.curectnnb1.org/.

**The paper explained**

**Problem**

CTNNB1 syndrome is a rare monogenetic neurodevelopmental disorder with no current treatment. It is characterized by cognitive and motor disabilities that impact the quality of life for these individuals.

**Results**

We have generated a Ctnnb1 germline heterozygous mouse line that displays key features of CTNNB1 syndrome in humans. We have identified novel molecular and functional changes in the brain and a small-molecule treatment that significantly normalizes the molecular, functional, learning, and motor phenotypes in this preclinical in vivo mouse model.

**Impact**

Our studies provide the first evidence for an efficacious treatment with the potential to significantly correct the phenotypes of CTNNB1 syndrome.

# Data availability

This study includes no data deposited to external repositories. Materials will be available upon written request and upon completion of the necessary Materials Transfer Agreement.

The source data of this paper are collected in the following database record: biostudies:S-SCDT-10_1038-S44321-024-00110-5.

# Peer review information

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

## Acknowledgements

The authors would like to acknowledge the following funding sources: NIH NIMH R01MH106623, NINDS R21NS119958-01A1 and Advancing CTNNB1 Cures and Treatments, Inc. (currently CTNNB1 Connect and Cure) (to MHJ). The authors would also like to acknowledge the Tufts Center for Neuroscience Research Behavior Core and Genomics Core with assistance with behavior and qPCR data.

## Author contributions

**Jonathan M Alexander**: Conceptualization; Data curation; Formal analysis; Supervision; Validation; Investigation; Visualization; Methodology; Writing—original draft; Project administration; Writing—review and editing. **Leeanne Vazquez-Ramirez**: Formal analysis; Validation; Investigation. **Crystal Lin**: Formal analysis; Validation; Investigation. **Pantelis Antonoudiou**: Data curation; Software; Formal analysis. **Jamie Maguire**: Supervision; Investigation; Methodology; Writing—original draft. **Florence Wagner**: Resources; Writing—review and editing. **Michele H Jacob**: Conceptualization; Resources; Supervision; Funding acquisition; Methodology; Writing—original draft; Project administration; Writing—review and editing.

Source data underlying figure panels in this paper may have individual authorship assigned. Where available, figure panel/source data authorship is listed in the following database record: biostudies:S-SCDT-10_1038-S44321-024-00110-5.

## Disclosure and competing interests statement

JMA, MHJ, and FFW are inventors on a GSK3-related patent application filed by Tufts University and the Broad Institute. FFW previously consulted for a biotechnology company on a GSK3-related project and is an inventor on multiple GSK3-related patent applications filed by the Broad Institute.

# Expanded View Figures

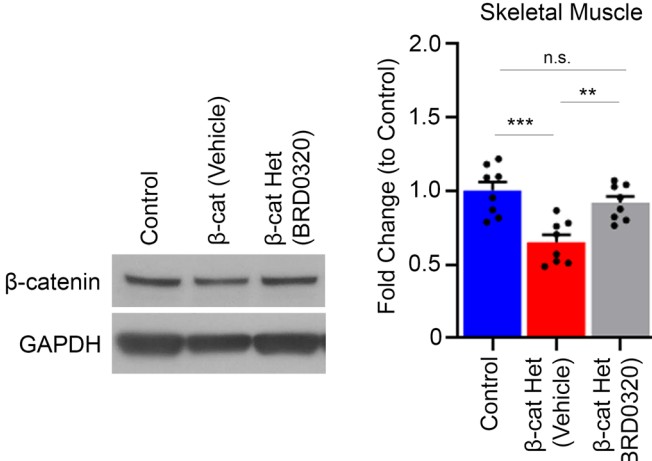

**Figure EV1.   Reduced β-catenin levels in skeletal muscle of β-cat mice are normalized by dual inhibition of GSK3α,β.**

Immunoblots and quantification show reduced protein levels of β-catenin in skeletal muscle (diaphragm) of β-cat het mice that can are increased to levels not significantly different from wild-type littermates by treatment with GSK3α,β dual inhibitor ($n = 8$ per genotype/condition; one-way ANOVA $F[2, 21] = 12.85$, $P < 0.001$; Tukey's multiple comparison $t$ test Control vs. vehicle-treated β-cat het, ***$P < 0.001$, Control vs. β-cat het dual inhibitor, **$P = 0.004$, β-cat het vehicle vs. β-cat het dual inhibitor, $P = 0.507$). All values are reported as mean of biological replicates $+/-$ s.e.m. from two independent experiments. Source data are available online for this figure.

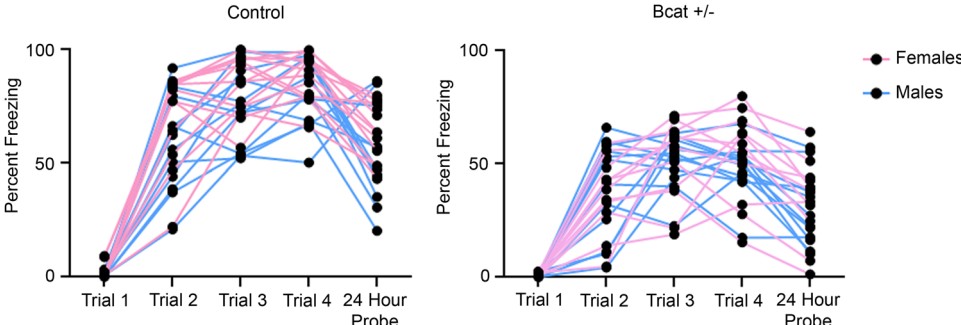

**Figure EV2. Individual mouse freezing behavior in the contextual fear conditioning task plotted across all training and the 24-hour probe trials show no single β-cat het mouse performed at consistently lowest levels and no sex difference in the cognitive capabilities.**

Control mice- $n = 13$ females, 12 males; repeated-measures ANOVA $F[1,23] = 7.006$ $P = 0.014$, Bonferroni's corrected $t$ test Trial 1: $P > 0.999$; Trial 2: $P = 0.531$; Trial 3: $P = 0.586$; Trial 4: $P = 0.736$; 24 h probe: $P = 0.443$); β-cat het mice- $n = 11$ females, 12 males; repeated-measures ANOVA $F[1,21] = 0.136$ $P = 0.716$, Bonferroni's corrected $t$ test Trial 1: $P > 0.999$; Trial 2: $P > 0.999$; Trial 3: $P > 0.999$; Trial 4: $P > 0.999$; 24 h probe: $P > 0.999$). All values are reported as mean of biological replicates $+/-$ s.e.m. from two independent experiments. Source data are available online for this figure.

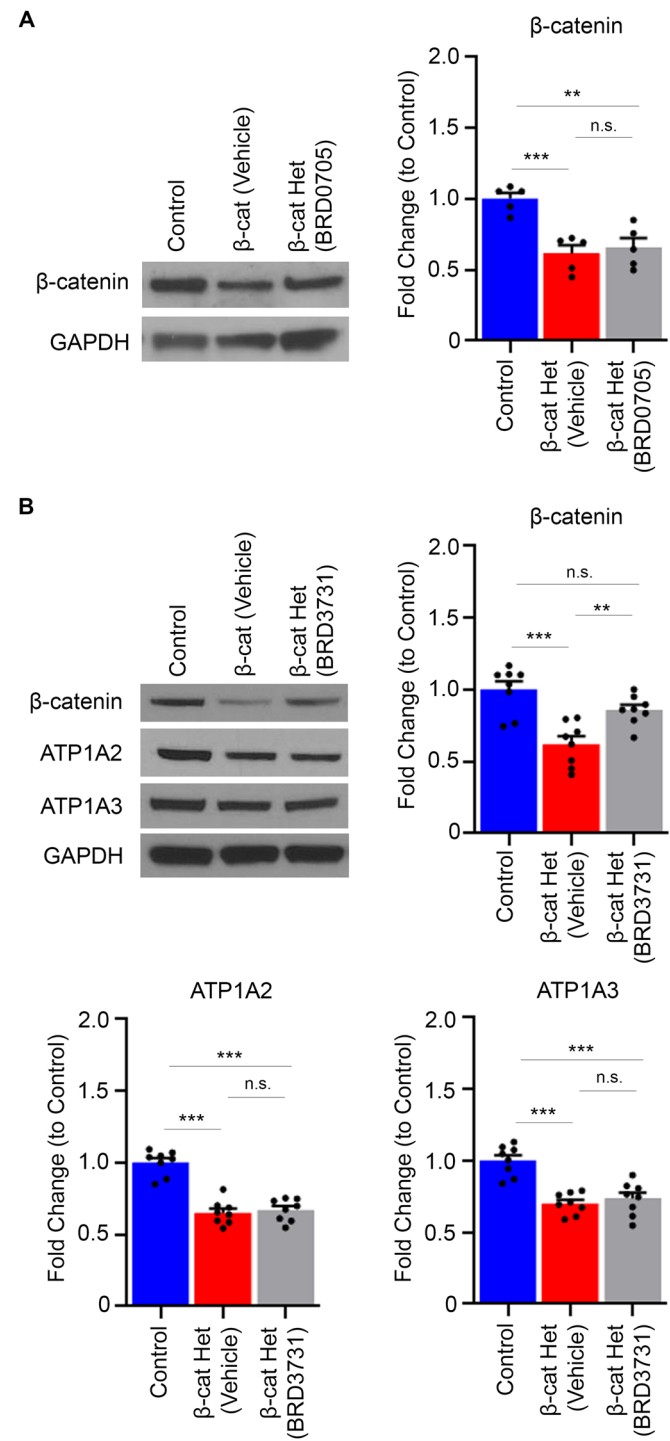

**Figure EV3. GSK3α and GSK3β selective inhibitors do not rescue molecular changes to the same extent as inhibition of both isoforms.**

(A) Immunoblots and quantification show that selective inhibition of the GSK3α isoform alone does not alter the reduced β-catenin protein levels in the β-cat het hippocampus, relative to vehicle-treated β-cat hets ($n = 5$ per genotype/condition; one-way ANOVA $F[2, 12] = 14.57$, $P < 0.001$; Tukey's multiple correction $t$ test Control vs. vehicle-treated β-cat, ***$P < 0.001$, Control vs. β-cat het GSK3α inhibitor, **$P = 0.002$, β-cat het vehicle vs. β-cat het GSK3α inhibitor, $P = 0.856$). (B) Selective inhibition of the GSK3β isoform significantly increases β-catenin protein levels in the hippocampus, relative to vehicle-treated β-cat hets, wherease it fails to significantly increase the reduced levels of the Na/K ATPase α2 and α3 isoforms ($n = 8$ per genotype/condition; β-catenin: one-way ANOVA $F[2, 21] = 14.89$, $P < 0.001$; Tukey's multiple comparison $t$ test Control vs. vehicle-treated β-cat het, ***$P < 0.001$, Control vs. β-cat het GSK3β inhibitor, $P = 0.127$, β-cat het vehicle vs. β-cat het GSK3β inhibitor, **$P = 0.008$; α2: one-way ANOVA $F[2, 21] = 44.78$, $P < 0.001$; Tukey's multiple comparison $t$ test Control vs. vehicle-treated β-cat het, ***$P < 0.001$, Control vs. β-cat het GSK3β inhibitor, ***$P < 0.001$, β-cat het vehicle vs. β-cat het GSK3β inhibitor, $P = 0.898$; α3: one-way ANOVA $F[2, 21] = 21.35$, $P < 0.001$; Tukey's multiple comparison t test Control vs. vehicle-treated β-cat het, ***$P < 0.001$, Control vs. β-cat het GSK3β inhibitor, ***$P < 0.001$, β-cat het vehicle vs. β-cat het GSK3β inhibitor, $P = 0.776$). All values are reported as mean of biological replicates $+/-$ s.e.m. from two independent experiments. Source data are available online for this figure.

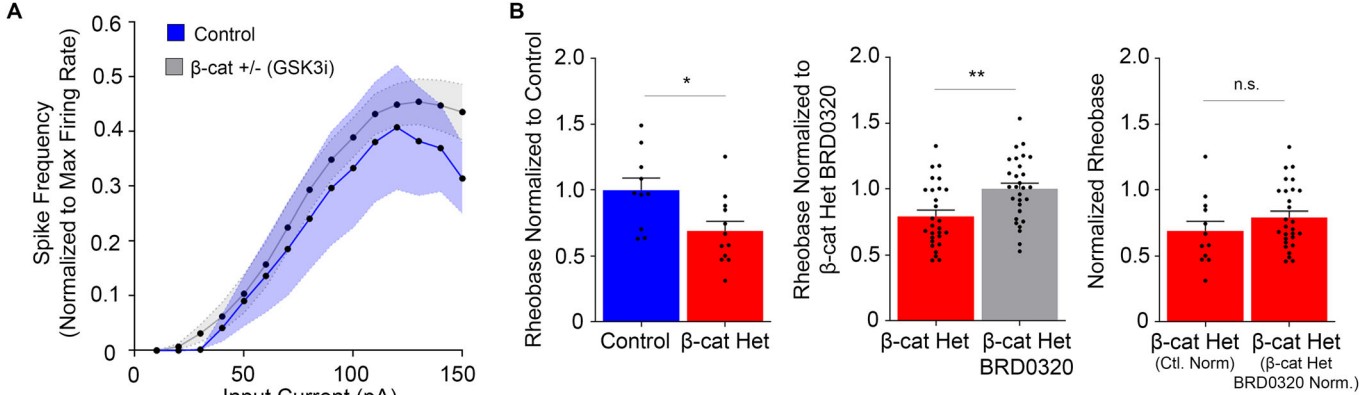

**Figure EV4. GSK3α,β dual inhibitor improves excitability of β-cat het mice to resemble control levels.**

(A) β-cat het mice treated with GSK3α,β dual inhibitor display spike frequency not significantly different from control littermates ($n = 10$ Controls, $n = 29$ β-cat het; repeated-measures ANOVA $F[1, 37] = 0.8189$, $P = 0.371$). (B) Histogram comparisons of the improved rheobase of β-cat hets treated with GSK3 dual inhibitor relative to vehicle-treated hets and wild-type littermates. Rheobases were normalized to either control (left panel: $n = 10$ Control, $n = 12$ β-cat het; Student's $t$ test $P = 0.016$) or GSK3α,β inhibitor-treated β-cat hets (center panel: $n = 28$ β-cat het vehicle-treated, $n = 29$ β-cat het GSK3i-treated; Student's $t$ test $P = 0.002$). Vehicle-treated β-cat hets showed significant decreases, to a similar extent, between the two normalizations suggesting no significant difference in excitability of GSK3 dual inhibitor hets compared to wild-types (right panel: $n = 12$ β-cat het normalized to Control, $n = 28$ β-cat het normalized to GSK3i-treated β-cat het; Student's $t$ test $P = 0.224$). All values are reported as mean of biological replicates $+/-$ s.e.m. from two independent experiments. Source data are available online for this figure.

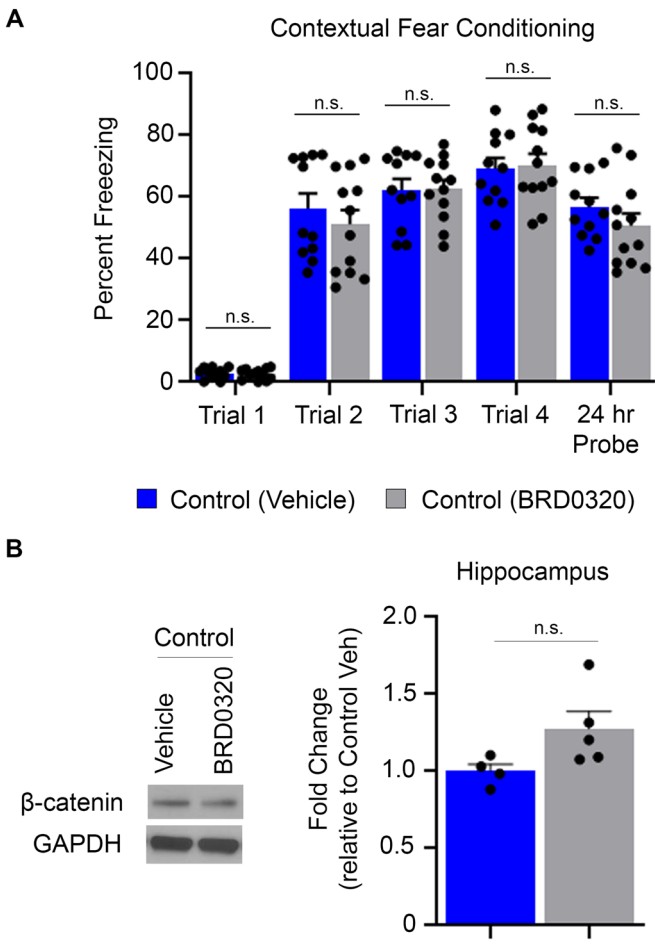

**Figure EV5. GSK3α,β dual-paralog inhibitor does not alter learning or β-catenin levels in wild-type mice.**

(A) Dual-paralog inhibition of GSK3α,β in wild-type mice does not significantly change their freezing behavior in the training trials and 24 h probe test in the contextual fear conditioning task, compared to vehicle-treated wild-type mice ($n = 11$ Controls vehicle-treated, 12 Controls dual inhibitor-treated; repeated-measures ANOVA $F[1,21] = 0.7150$, $P = 0.407$). (B) Further, it does not significantly alter β-cat protein levels in the hippocampus ($n = 4$ Controls vehicle-treated, 5 Controls dual inhibitor-treated; Student's $t$ test, $P = 0.578$). All values are reported as mean of biological replicates $+/-$ s.e.m. from two independent experiments. Source data are available online for this figure.

## Contextual Fear Condition (GSK3β Inhibitor)

**Figure EV6. GSK3β selective inhibitor does not correct cognitive deficits of β-cat het mice to the same extent as the GSK3α,β dual inhibitor.**

Inhibition of GSK3β alone shows a modest significant improvement in freezing behavior at trial 4 that is not retained in the 24 h probe trial ($n = 13$ Controls, $n = 11$ vehicle-treated β-cat het, $n = 11$ GSK3β inhibitor-treated β-cat het; repeated-measures ANOVA $F[2, 32] = 73.32$, $P < 0.001$; Tukey's multiple comparison $t$ test Trial 1: Control vs. β-cat het vehicle, $P = 0.586$; Control vs. β-cat het GSK3β inhibitor, $P = 0.671$, β-cat het vehicle vs. β-cat het GSK3β inhibitor, $P = 0.122$; Trial 2: Control vs. β-cat het vehicle, $P = 0.085$; Control vs. β-cat het GSK3β inhibitor, $P = 0.384$, β-cat het vehicle vs. β-cat het GSK3β inhibitor, $P = 0.843$; Trial 3: Control vs. β-cat het vehicle, ***$P < 0.001$; Control vs. β-cat het GSK3β inhibitor, **$P = 0.002$, β-cat het vehicle vs. β-cat het GSK3β inhibitor, $P = 9.37$; Trial 4: Control vs. β-cat het vehicle, ***$P < 0.001$; Control vs. β-cat het GSK3β inhibitor, *$P < 0.001$, β-cat het vehicle vs. β-cat het GSK3β inhibitor, **$P = 0.006$; 24 h Probe: Control vs. β-cat het vehicle, ***$P < 0.001$; Control vs. β-cat het GSK3β inhibitor, ***$P < 0.001$, β-cat het vehicle vs. β-cat het GSK3β inhibitor, $P = 0.844$). All values are reported as mean of biological replicates $+/-$ s.e.m. from two independent experiments. Source data are available online for this figure.

