## [Peer Review File · EMBO Molecular Medicine]

Inhibition of GSK3 α,β rescues cognitive phenotypes in a preclinical mouse model of CTNNB1 Syndrome

Jonathan Alexander, Leeanne Vazquez-Ramirez, Crystal Lin, Pantelis Antonoudiou, Jamie Maguire, Florence Wagner, and Michele Jacob

Corresponding author: Michele Jacob (michele.jacob@tufts.edu)

Review Timeline:

Submission Date:	18th Jul 23
Editorial Decision:	14th Aug 23
Appeal:	23rd Aug 23
Editorial Decision:	24th Aug 23
Revision Received:	11th May 24
Editorial Decision:	10th Jun 24
Revision Received:	25th Jun 24
Editorial Decision:	5th Jul 24
Revision Received:	12th Jul 24
Accepted:	17th Jul 24

Editor: Poonam Bheda

Transaction Report:

11th Aug 2023

Decision on your manuscript EMM-2023-18356-T

Dear Dr. Jacob,

Thank you for the submission of your manuscript "Inhibition of GSK3 α/β rescues cognitive phenotypes in a preclinical mouse model of CTNNB1 Syndrome". We have now received feedback from the three referees who had agreed to review your manuscript.

As you will see from the reports below, the referees acknowledge the potential interest of the study, however they also mention several shortcomings (including, but not limited to, discrepancies and issues of sample size, sex, age, statistics, and potential inconsistencies with previous work, also to ensure that the outliers are not causing the results, rescue experiments do not seem to be done with age-matched mice, and the use of a whole-body versus forebrain-specific knockout). Based on the referees' reports, I am afraid I see little choice but to return the manuscript to you at this point with the decision that we cannot offer to publish it.

While we cannot pursue this manuscript further, we encourage you to transfer your study to our not-for-profit open-access sister journal, Life Science Alliance (LSA). We shared your manuscript and the accompanying reviews with LSA Executive Editor, Eric Sawey, who is interested in these findings, and would like to invite further consideration of this manuscript at LSA pending the following revisions:

- Address Reviewer 1's Primary Concerns #1, 2 & 4 via added discussion. Minor Concerns #1, 3 & 4 should be addressed.
- Address Reviewer 2's Points #1, 3, 4 & 6, and the Minor points.
- Address Reviewer 3's comments.

We understand that such a revision might need to be re-reviewed, in which case, Dr. Sawey will walk the Reviewers through our transfer process.

We encourage you to use the link below to transfer your manuscript to LSA. You do not need to revise the manuscript before transferring it to LSA. Once you transfer, Dr. Sawey will email you an invitation to revise and resubmit, listing the same revision requests as mentioned above. Please feel free to reach out at e.sawey@life-science-alliance.org if you have any questions about the LSA journal, the transfer process or the revisions requested.

I am very sorry to disappoint you on this occasion and I hope you will view the possibility of a transfer favorably. If this is the case, please use the link below to transfer the manuscript directly.

With kind regards,

Poonam Bheda

Poonam Bheda, PhD
Scientific Editor
EMBO Molecular Medicine

***** Reviewer's comments *****

Referee #1 (Comments on Novelty/Model System for Author):

1. The employment of whole-body knockout models prompts inquiry. Notably, numerous publications have detailed Ctnnb1 knockdown in both whole-body and forebrain contexts. Given that whole-body knockout impacts not just the central nervous system but other systems as well, concurrent disruption of locomotor behaviors is observed in this study. It is imperative to elucidate the rationale behind selecting this whole-body knockout model and discern its advantages v.s. forebrain knockdown.
2. Discrepancies between the current study and a previous publication (Wickham et al., 2019) are evident in relation to N-cadherin and DKK-1 expression. Notably, the present study shows an unaltered expression of N-cadherin and α -N cadherin, coupled with DKK-1 downregulation, in contrast to the earlier findings. A cogent explanation is warranted to reconcile these discrepancies.

Referee #1 (Remarks for Author):

Alexander et al. have reported the discovery of a high-selectivity dual inhibitor targeting GSK3 α/β , which demonstrates the normalization of β -catenin and Na/K ATPase levels. This normalization brings these levels in line with the baseline ranges observed in wildtype littermates. Furthermore, this inhibitor showcases the potential to ameliorate cognitive impairments evident in whole-body Ctnnb1 knockout mice. While the findings offer intriguing insights, their novelty appears somewhat restricted.

Primary Concerns:

1. The employment of whole-body knockout models prompts inquiry. Notably, numerous publications have detailed Ctnnb1 knockdown in both whole-body and forebrain contexts. Given that whole-body knockout impacts not just the central nervous system but other systems as well, concurrent disruption of locomotor behaviors is observed in this study. It is imperative to elucidate the rationale behind selecting this whole-body knockout model and discern its advantages v.s. forebrain knockdown.
2. Discrepancies between the current study and a previous publication (Wickham et al., 2019) are evident in relation to N-cadherin and DKK-1 expression. Notably, the present study shows an unaltered expression of N-cadherin and α -N cadherin, coupled with DKK-1 downregulation, in contrast to the earlier findings. A cogent explanation is warranted to reconcile these discrepancies.
3. While past research by the same group has delineated synaptic changes stemming from both whole-body and forebrain Ctnnb1 knockdown, however, the current study deviates by focusing on neuronal excitability. Suppressed action potential firing rates and afterhyperpolarization were observed. Although Na/K ATPase was chosen to elucidate neuronal activity alterations, consideration should be given to the potential involvement of the HCN channel. Moreover, the mechanisms underlying suppressed Na/K ATPase function consequent to Ctnnb1 knockdown remain unexplored, as do the mechanisms governing the regulatory influence of GSK3 α/β inhibitor on Na/K ATPase.
4. The differential effects of β -selective and α -selective GSK3 inhibitors on β -catenin levels prompt inquiry. Specifically, the β -selective inhibitor does not engender significant molecular changes or behavioral effects, while the dual inhibition of both GSK3 α and GSK3 β appears consequential. An explicit elucidation of the mechanisms dictating these disparate outcomes is necessary. Additionally, the absence of treatment effects with the GSK3 β inhibitor alone warrants clarification.

Minor Concerns:

1. Given the extensive focus on the hippocampus within this article, a rationale for not conducting behavioral tests relating to spatial and learning memory but via fear conditioning is warranted. Fear conditioning is a behavioral assay often employed to probe animal learning and memory particularly in the context of fear. Given the role of the amygdala in fear-related processes, it is pertinent to consider whether similar Ctnnb1 downregulation is observed in the amygdala of β -cat $+/$ - mice.
2. The cognitive and motor impairments evident in β -cat $+/$ - mice underscore the relevance of assessing forelimb grip strength and rotarod performance. These evaluations could provide insights into potential enhancements facilitated by the dual GSK3 α and β inhibitor.
3. Notably absent from the study is data pertaining to the neuronal excitability of β -cat $+/$ - post-treatment with the GSK3 inhibitor. Furthermore, data characterizing molecular changes and behavioral outcomes subsequent to GSK3 inhibitor administration in β -cat $+/$ - mice are lacking.
4. Cytosol and nucleus Ctnnb1 is upregulated but suppressed synapse Ctnnb1 in Shank3 mice. But not as simply stated as down-regulated Ctnnb1 in Shank3 mice.

Referee #2 (Comments on Novelty/Model System for Author):

The statistical tests are fine overall but there are some outliers restricted to the experimental animals that may drive the result. Some controls are missing. The mouse models themselves are not novel (have been previously generated) but the examination of them is. Medical impact is difficult to assess as only a single dose of inhibitor was given, no longer term effects measured.

Referee #2 (Remarks for Author):

Here, the authors have generated a heterozygous mouse that lacks one copy of beta-catenin. The rationale was to mimic a human disorder observed relatively recently where inactivation of the human cognate occurs and that is associated with significant behavioural deficits. The haploinsufficient animals demonstrate certain behavioural changes compared to littermates that may relate to the patients phenotype and the authors show that these can be rescued by administration of a pan-isoform GSK3 inhibitor. This may have future implication for potential therapy.

1. It is perhaps surprising that there is a similar deficit in beta-catenin protein as RNA (which is expected) as shown in figure 1. Typically, the majority of beta-catenin in cells is associated with the cadherins and excess is rapidly degraded via ubiquitin-mediated proteolysis via the destruction complex. The authors should estimate not only total beta-catenin but soluble (non-cadherin-associated beta-catenin) in addition to levels of N-cadherin and N-cadherin associated beta-catenin as they have done (in figure 3B). Ideally, this should be performed in the brain and at least one other organ (e.g. E-cadherin), given the whole body knockout. In this respect, the emphasis is almost exclusively on the brain, when there may be inter-organ aspects.

2. The fear conditioning test should be supported by other short term memory tests such as Y maze. In general, the authors should perform an array of behavioural tests to assess the cognitive and other functions. As it stands, the data is rather minimal with respect to comparison with the human disorder.

3. As alluded to above, the Wnt pathway is well known for robust compensation. Changes in levels of components tend not to have large effects on output due to hysteresis feedback where several Wnt targets act to shut down signaling - e.g Axin 2 expression and FrzB hence limits to throughput are adapted. Plakoglobin has also been shown to be induced when beta-catenin is reduced. The authors have measured DKK1 and En2 but the signalling picture would be clearer with measurement of Axin2 and FrzB.

4. The GSK3 inhibitor data in figure 6 also raise some questions regarding levels of beta-catenin. They are observed to rise upon drug treatment but it is not shown whether this is the soluble and/or cadherin-associated beta-catenin. Given the majority of beta-catenin, as noted above, is cadherin-associated, inhibition of the kinase would not be expected to have a large effect on total beta-catenin. As in point 1, it is important to measure both populations (cadherin-associated and soluble).

5. As mentioned in point 3, the effects of the inhibitors should include levels of Axin2 and/or FrzB to directly measure output of the Wnt pathway. This would also help address the important question of how much inhibition is required to compensate for the gene loss of beta-catenin given the authors are interested in possible therapeutic strategies. Given the isoform selective inhibitors individually do not reverse the impact of loss of the gene, this may suggest a high degree of inhibition is needed. If so, this may be problematic due to many other targets of the kinases.

6. Na/K ATPase alpha1 protein should be included in figure 6.

Minor points:

A. Both sexes were used in the behavioural experiments but were sufficient animals assessed to determine whether there were statistically significant differences based on sex?

B. The distribution of individual animal results in figure 2A is unusual (note the +/- animals include several with very low freeze rates. Were these animals sick (there are reports of other effects of gene reduction of beta-catenin)? Is the statistical difference driven by these animals?

Referee #3 (Comments on Novelty/Model System for Author):

More information on the natural history of disease progression in this animal model is required for evaluating the efficacy of corrective treatment at pre-symptomatic or symptomatic stage.

Referee #3 (Remarks for Author):

On the premise that CTNNB1 syndrome is commonly associated with variants of CTNNB1 resulting in haploinsufficiency, the disease modelling study was undertaken in a heterozygous loss-of-function model. The findings point to the disruption in Na/K ATPase and synaptic adhesion are potential pathophysiological factors that underpin the altered electrochemical activity of cortical and hippocampal neurons in CTNNB1-related disease manifesting neurological dysfunction. By normalizing the level of CTNNB1 in the mutant animal and thereby restoring Na/K ATPase activity, through inhibiting glycogen synthase kinase activity, the molecular and neurological phenotypes can be reverted.

Points for clarification

Behavioral assessment:

- For the contextual fear conditioning test and the rotarod test, clarify which datasets that were analyzed by multiple t-test, versus

those by ANOVA, and explain which of these two types of result of statistical significance is relevant to the interpretation of the outcome of the behavioral test. How was the percent freezing determined (Fig 2A, 7) - did some mutant mice not display any defect in contextual fear conditioning?

- For the forelimb grip-strength test, clarify the discrepancy in the reported p-value ($p=0.003$ average of 3 trials - page 6 paragraph 2) and the figure ($* p<0.001$) to ensure accuracy in the statistical analysis.
- For the open field test, clarify whether other parameters, such as total distance travelled or distance travelled in the center, were measured in the open field. These additional parameters are relevant in assessing the animals' exploratory and anxiety-related behaviors, which are of interest to establish CTNNB1-like phenotype, and their inclusion could strengthen the study's findings.

Functional property of hippocampal neuron:

- It was indicated that the assays of excitability were performed on CA1 pyramidal neuron and dentate gyrus granule cells (DGGCs) (page 8 paragraph 2), but only the data of DGGCs were presented in Figure 4. What were the data ("similar functional changes" - figure legend and page 8 paragraph 2) of the CA1 pyramidal neurons omitted? Levels of the three Na/K ATPase catalytic units were shown for the cortex and hippocampus, but not the "frontal cortex" (Fig. 5). Which part/s of the "cortex" that was/were outside the "frontal cortex" were studied? Why were the "similar" frontal cortex data not shown? "Data not shown" is not acceptable if the data were an integral part of results, especially that the impact of GSK3 inhibition was evaluated on "undefined" parts (or neuronal types) in the frontal cortex and hippocampus (Fig. 6)

GSK3 Rescue:

- It was noted that "for all experiments 2-3-month-old mice of both sexes were used" (materials and methods), i.e., 8-12-week-old mice were used. Instead, the rescue experiments were performed on 6-weeks old mice (figure 6 legend). How was this timing chosen? No information of the natural history of cognitive and motor defect in this model was provided to indicate whether the mutant mice were symptomatic or pre-symptomatic for behavioral defect and neuronal dysfunction and if the phenotype is fully penetrant. This is a critical point to consider in the light of the assertion (Discussion) of significant corrective outcome with treatment "at young adult ages when the phenotypes are clearly displayed" (page 18) and that "These findings suggest a relatively wide age window for therapeutic intervention by showing that treatment initiated after the onset of symptoms still provides significant corrective outcomes" (page 15).

Specific points

Indicate the age of the animals analyzed for behavioral phenotypes, and those from which tissues were sampled for biochemistry studies (materials and methods).

Text:

- The labels of the panels (in lower case) did not match those in the legend (in capitals)
- Clarify whether the correct p-value is $p<0.01$ or $p<0.001$ (page 6, line 8) for over the training trial results to ensure accuracy in the statistical analysis.
- It would be appropriate to (i) revise the section heading to: "... improves the cognitive function of b-cat+/- mice" (page 12) and (ii) present the statistical results in the figure 7 legend or in tabulated form in the Fig 7, and not in the text (page 13 paragraph 2).
- The sentence following "In summary" (page 19) is a re-iteration of the introduction and does not belong to the summary.

Figures:

- Fig. 1 and 2 Address the inconsistency in the reported p-value on page 21, line 17, and page 6, regarding the statistical significance. The correct and consistent p-value should be provided to avoid any potential misinterpretation of the study results.
- Fig. 2: It is essential to explain why there is a discrepancy in the number of mice shown in the graphs of Figure 2 if the tests were performed on the same animals. This inconsistency might impact the reliability of the results, and further clarification is needed. Fig 2D: The inconsistency in the reported sample size should be addressed. If the control group indeed consists of 9 values, the correct sample size should be stated accurately as "n=9" on page 6 last line to avoid confusion.
- Fig. 3: Clarify (i) which of the double protein bands of N-cadherin and LEF1 were subjected to quantification (Fig. 3A), (ii) the apparent inconsistency of the IP data (Fig. 3B) and the quantified results, and (iii) the discrepancy of the quantified data with the blot data of DKK1 protein (Fig. 3C).
- Fig. 4: Explain the discrepancy between the stated sample size on page 8 and the number of data points shown in Figure 4. If the graphs display more than eight (8) values per group (or $n=4$ independent samples with 2 replicates each), the authors should clarify the reason for the inclusion of more data points in the figure to ensure the accuracy of the data presentation. For Fig. 4B, and 4E: Provide the colour legend of the two tracings.
- Fig. 2A, 6 and 7: Indicate in the figures which pairs of datasets were assessed (e.g., Supplementary figure 1A) to reach the respective statistical results.

As a service to authors, EMBO provides authors with the possibility to transfer a manuscript that one journal cannot offer to publish to another EMBO publication. The full manuscript and if applicable, reviewers reports are automatically sent to the

receiving journal to allow for fast handling and a prompt decision on your manuscript. For more details of this service, and to transfer your manuscript to another EMBO title please click on Link Not Available

Dear Dr Bheda,

We have now considered the reviewers' critiques in detail and request that you reconsider the rejection of this manuscript (EMM 2023-18356-T). The reviewers' primary concerns are ill-founded and can be readily addressed as discussed below. Some of the comments are helpful and motivate us to add a few additional experiments to further clarify b-catenin intracellular distributions in the heterozygotes, and measure additional Wnt gene expression changes. This will expand the data but not alter the conclusions of the study. As questioned by reviewer 2, there are no outliers in any of the data sets that could drive the results. All data is based on quantitative assessments in PRISM and includes ROUT tests to identify outliers. None were detected. No mice were sick, our veterinary staff provide daily health checks. As far as reviewer 3 questioning potential age mismatches for rescue experiments, all experiments in this study used mice from 6 weeks to 12 weeks of age and every experiment included littermate controls, to define the phenotype of heterozygotes compared to wildtype littermates, and to compare drug-treated heterozygotes to vehicle-treated heterozygotes and wildtypes, all littermates. As indicated in the text, 2-3 months of age is the standard for reliable quantitative assessment of complex cognitive tasks in mice. Further, we will correct the few typos that led to a small number of mismatches between the text, figures and figure legends (e.g., sample size greater in Fig.4 vs text p.8, differing significant p values). Below, we provide further details to support our request for you to reconsider the rejection decision.

Reviewer-1 did not appear to appreciate that the global CTNNB1 heterozygote mouse is designed to model the whole body CTNNB1 haploinsufficiency of the human disorder CTNNB1 Syndrome. The insights provided by our study have high translational significance for this whole-body heterozygote disorder. We identify novel molecular and functional pathophysiological changes in the brain and provide the first evidence for a corrective treatment. These data have high significance for this incapacitating human disease that is characterized by cognitive impairments, microcephaly and spastic gait. There are no current treatment options for these individuals.

Reviewer-1 concern that the data are inconsistent with our previous study is not well founded. Our previous study used a mouse line with conditional homozygous knockout of CTNNB1 targeted to forebrain excitatory neurons. The models differ in genetic manipulations, heterozygotes vs homozygotes, whole body vs excitatory neurons. They therefore differ, not unexpectedly, in outcomes. We will expand the discussion of these differences in the revised manuscript. Importantly, CTNNB1 conditional homozygous knockout in forebrain neurons does not model human disease, whereas whole body CTNNB1 heterozygosity does.

We plan to add some new experiments, as recommended by Reviewer-2. The new data will strengthen, but not alter the conclusions, such as more data on the changes in b-catenin interactions with N-cadherin and Wnt gene expression levels in the presence vs absence of the GSK3 inhibitor treatment.

Reviewer-3 asks us to explain when we analyzed the behavioral datasets by "multiple t-test versus ANOVA" and the "significance for interpreting the outcomes". For all experiments, when either more than two groups are involved or when the same mice are being evaluated over separate trials (contextual fear conditioning), we apply an ANOVA (either one-way ANOVA for a single comparison between more than two groups, or a repeated measure ANOVA to evaluate multiple groups of mice over repeated trials). When the ANOVA indicates a significant change, we apply multiple comparison t-tests to evaluate the significance of the group variability. We reported the ANOVA values in the text, and the multiple t-test values in the Fig. legends, possibly causing the confusion for the reviewer.

Some statistical comparisons they requested are presented in the figures. It is not clear what inconsistency the reviewer finds in Fig.3B and C. We will fix the few typos they found as mismatches

between the text and a few figure legends. We apologize for the confusion regarding frontal cortex versus cortex. We always used frontal cortex, but occasionally referred to the tissue as cortex. We will clarify this in the revised text.

We welcome your response and would be happy to discuss these issues in more detail.

Best,

Michele

24th Aug 2023

Dear Dr. Jacob,

Thank you for your response to the editorial decision on your manuscript entitled. I have now carefully examined the arguments provided in your letter and discussed them with the other members of our editorial team. We would be willing to reconsider your manuscript if you provide data that would considerably strengthen the message of the study and address the referees concerns in full. From your email it seems that you can address all of the referees concerns with clarifications and additional experimental evidence and analyses, so please submit the revised version of the manuscript within the next three months. Please let us know if you require longer to complete the revision. Further consideration of the revised article will entail a second round of review and we cannot guarantee the outcome of the reevaluation; therefore, I would strongly advise against returning an incomplete revision.

We require:

4) A .docx formatted letter INCLUDING the reviewers' reports and your detailed point-by-point responses to their comments. As part of the EMBO Press transparent editorial process, the point-by-point response is part of the Review Process File (RPF), which will be published alongside your paper.

5) A complete author checklist, which you can download from our author guidelines (<https://www.embopress.org/page/journal/17574684/authorguide#submissionofrevisions>). Please insert information in the checklist that is also reflected in the manuscript. The completed author checklist will also be part of the RPF.

6) Please note that all corresponding authors are required to supply an ORCID ID for their name upon submission of a revised manuscript.

7) It is mandatory to include a 'Data Availability' section after the Materials and Methods. Before submitting your revision, primary datasets produced in this study need to be deposited in an appropriate public database, and the accession numbers and database listed under 'Data Availability'. Please remember to provide a reviewer password if the datasets are not yet public (see <https://www.embopress.org/page/journal/17574684/authorguide#dataavailability>).

This study includes no data deposited in external repositories.

8) For data quantification: please specify the name of the statistical test used to generate error bars and P values, the number (n) of independent experiments (specify technical or biological replicates) underlying each data point and the test used to calculate p-values in each figure legend. The figure legends should contain a basic description of n, P and the test applied. Graphs must include a description of the bars and the error bars (s.d., s.e.m.). Please provide exact p values.

- For the figures that you do NOT wish to display as Expanded View figures, they should be bundled together with their legends

in a single PDF file called *Appendix*, which should start with a short Table of Content. Appendix figures should be referred to in the main text as: "Appendix Figure S1, Appendix Figure S2" etc.

13) Author contributions: CRediT has replaced the traditional author contributions section because it offers a systematic machine readable author contributions format that allows for more effective research assessment. Please remove the Authors Contributions from the manuscript and use the free text boxes beneath each contributing author's name in our system to add specific details on the author's contribution. More information is available in our guide to authors.

Please also suggest a striking image or visual abstract to illustrate your article as a PNG file 550 px wide x 300-600 px high. Share synopsis text and image, as well as eTOC:

Please note that these would be the final versions and changes during proofing are usually not allowed

16) As part of the EMBO Publications transparent editorial process initiative (see our Editorial at <http://embomolmed.embopress.org/content/2/9/329>), EMBO Molecular Medicine will publish online a Review Process File (RPF) to accompany accepted manuscripts.

In the event of acceptance, this file will be published in conjunction with your paper and will include the anonymous referee reports, your point-by-point response and all pertinent correspondence relating to the manuscript. Let us know whether you agree with the publication of the RPF and as here, if you want to remove or not any figures from it prior to publication. Please note that the Authors checklist will be published at the end of the RPF.

I look forward to receiving your revised manuscript.

Yours sincerely,

Poonam Bheda

Poonam Bheda, PhD
Scientific Editor
EMBO Molecular Medicine

Poonam Bheda. PhD
Scientific Editor
EMBO Molecular Medicine

Detailed response to the Reviewer's comments (Manuscript EMM-2023-18356-V2-Q)

We thank the reviewers for their thoughtful comments. In response, we have added text to clarify specific points and new data that strengthen the results and improve the manuscript. Below, we respond to each of the comments. As there are changes throughout the entire manuscript, they are not highlighted, rather we indicate the page number of all major changes in this response letter.

Referee #1 (Comments on Novelty/Model System for Author):

1. The employment of whole-body knockout models prompts inquirv. Notably, numerous publications have detailed Ctnnb1 knockdown in both whole-body and forebrain contexts. Given that whole-body knockout impacts not just the central nervous system but other systems as well, concurrent disruption of locomotor behaviors is observed in this study. It is imperative to elucidate the rationale behind selecting this whole-body knockout model and discern its advantages v.s. forebrain knockdown.

The *Ctnnb1* germline heterozygote mouse is designed to model the human disorder CTNNB1 syndrome caused by *CTNNB1 de novo* pathogenic heterozygous loss-of-function variants that result in haploinsufficiency. The insights provided by our study have high translational significance for this human heterozygote disorder. We show that the *Ctnnb1* germline heterozygous mouse displays cognitive and motor impairments, resembling key features of human CTNNB1 syndrome. We identify novel molecular and functional pathological changes in the brain. We provide the first evidence for an efficacious treatment that significantly normalizes the phenotypes in our preclinical *in vivo* mouse model. There are no current treatment options for this severe human developmental disorder, underscoring the significance of our study identifying a potential corrective treatment. We have expanded the text (Introduction p.4, Results p.6) to clarify this issue.

2. Discrepancies between the current study and a previous publication (Wickham et al., 2019) are evident in relation to N-cadherin and DKK-1 expression. Notably, the present study shows an unaltered expression of N-cadherin and α -N cadherin, coupled with DKK-1 downregulation, in contrast to the earlier findings. A cogent explanation is warranted to reconcile these discrepancies.

We feel that the data between the current study and our previous study (Wickham et al., 2019) are not inconsistent. The mouse models differ in genetic manipulations, *Ctnnb1* global heterozygotes vs homozygous conditional knockout of *Ctnnb1* targeted to forebrain excitatory neurons. Importantly, *Ctnnb1* germline heterozygosity models the human disease CTNNB1 syndrome, whereas *Ctnnb1* homozygous conditional knockout in forebrain neurons does not. The two mouse models differ in outcomes, displaying significantly different levels of β -catenin, the protein encoded by the *Ctnnb1* gene, and impact on β -catenin functions. *Ctnnb1*/ β -catenin mRNA and protein levels are reduced approximately 50% in the heterozygotes vs complete loss in the homozygote forebrain neurons which, in turn, leads to upregulation of the partial functional homolog γ -catenin. The upregulated γ -catenin compensates for the loss of β -catenin in the Wnt signal transduction pathway, but only partially compensates for β -catenin functions in the N-cadherin based synaptic adhesion complex due to differences in protein structure between the two catenins, as detailed in the Wickham et al, 2019 study. We have added text (first paragraph of the Results, p.6) that states the difference in the genetic manipulations that underlie the different outcomes between our current and past studies.

Referee #1 (Remarks for Author):

Alexander et al. have reported the discovery of a high-selectivity dual inhibitor targeting GSK3 α/β , which demonstrates the normalization of β -catenin and Na/K ATPase levels. This normalization brings these levels in line with the baseline ranges observed in wildtype littermates. Furthermore, this inhibitor showcases the potential to ameliorate cognitive impairments evident in whole-body Ctnnb1 knockout mice. While the findings offer intriguing insights, their novelty appears somewhat restricted.

The novelty and significance of our findings are the identification of novel molecular and functional pathophysiological changes in the brain, *in vivo*, caused by Ctnnb1/ β -catenin haploinsufficiency and the first evidence for an efficacious therapeutic treatment that significantly normalizes the phenotypes. Further, our study takes advantage of the latest advances by using a newly developed GSK3 α/β inhibitor that exhibits exquisite specificity and good brain permeability, compared to other current GSK3 inhibitors. Our data have high translational impact as CTNNB1 syndrome lacks any current treatment and the disabilities severely impact the quality of life for these individuals. Moreover, several other human gene mutations also cause reduced β -catenin levels/functions and similar developmental disorders. These disorders may share a common pathology, at least in part, and benefit from the therapeutic strategies that we are developing for CTNNB1 syndrome.

Primary Concerns:

1. The employment of whole-body knockout models prompts inquiry. Notably, numerous publications have detailed Ctnnb1 knockdown in both whole-body and forebrain contexts. Given that whole-body knockout impacts not just the central nervous system but other systems as well, concurrent disruption of locomotor behaviors is observed in this study. It is imperative to elucidate the rationale behind selecting this whole-body knockout model and discern its advantages v.s. forebrain knockdown.

Same as comment 1 above; see response above.

2. Discrepancies between the current study and a previous publication (Wickham et al., 2019) are evident in relation to N-cadherin and DKK-1 expression. Notably, the present study shows an unaltered expression of N-cadherin and α -N cadherin, coupled with DKK-1 downregulation, in contrast to the earlier findings. A cogent explanation is warranted to reconcile these discrepancies.

Same as comment 2 above; see response above.

3. While past research by the same group has delineated synaptic changes stemming from both whole-body and forebrain Ctnnb1 knockdown, however, the current study deviates by focusing on neuronal excitability.

This report is our first study of whole-body Ctnnb1/ β -catenin reduction by germline heterozygosity. Our previous published study was CamKII-Cre-mediated conditional knockout (homozygotes) of Ctnnb1/ β -catenin in mouse forebrain excitatory neurons. We assess neuronal excitability in the current study of the Ctnnb1 heterozygote mouse brain to begin to define functional changes that are likely relevant to pathophysiological changes in the brain of individuals with CTNNB1 syndrome.

Suppressed action potential firing rates and afterhyperpolarization were observed. Although Na/K ATPase was chosen to elucidate neuronal activity alterations, consideration should be given to the potential involvement of the HCN channel.

We agree that, in addition to the Na/K ATPase changes we identify, other channels such as HCN may also play a role in the functional changes. We have added a statement to this effect (p. 10) and propose this as a future direction (p.20).

*Moreover, the mechanisms underlying suppressed Na/K ATPase function consequent to *Cttnb1* knockdown remain unexplored, as do the mechanisms governing the regulatory influence of GSK3 α / β inhibitor on Na/K ATPase.*

Previous studies show that the Na/K ATPase α 3 associates with the N-cadherin, β -catenin synaptic adhesion complex in cultured hippocampal neurons (Tanaka et al., 2012; added to p.10, 18). Studies of C2C12 muscle cell lines show similar changes between Na/K ATPase α 2 and β -catenin levels, both down- and up-regulation (via β -catenin siRNA knockdown to 50% levels and overexpression; Zhao et al., 2014, 2019), as mentioned previously and expanded in the revised manuscript (p.10, 18). We show here in the mouse brain *in vivo*, similar changes between Na/K ATPases α 2 and α 3 with β -catenin levels, both down and up (reduced in heterozygotes and normalized in GSK3 inhibitor treated heterozygotes; Fig.6). We plan future studies to uncover the underlying regulatory mechanisms and to identify additional changes in functional and synaptic proteins caused by β -catenin haploinsufficiency.

4. The differential effects of β -selective and α -selective GSK3 inhibitors on β -catenin levels prompt inquiry. Specifically, the β -selective inhibitor does not engender significant molecular changes or behavioral effects, while the dual inhibition of both GSK3 α and GSK3 β appears consequential. An explicit elucidation of the mechanisms dictating these disparate outcomes is necessary. Additionally, the absence of treatment effects with the GSK3 β inhibitor alone warrants clarification.

GSK3 α , β paralogs are derived from different genes, and have both redundant and unique substrates and functions. Precedence shows that inhibition of both GSK3 α and β are required to increase β -catenin levels in the mouse and in human cell lines *in vitro* (Doble et al., 2007; Wagner et al., 2018; McCamphill et al., 2020). For example, increasing β -catenin levels requires silencing 3 of the 4 GSK3 α and β alleles in mouse embryonic stem cells (Doble et al., 2007). We obtained similar findings in our *in vivo* *Cttnb1* het mouse study. The highly selective GSK3 α , β dual paralog inhibitor significantly normalized β -catenin levels and phenotypes of the hets, whereas the GSK3 α -selective and β -selective inhibitors were less effective in the same treatment paradigm. The α -selective inhibitor treated hets showed no significant changes relative to vehicle-treated het littermates. The β -selective inhibitor was partially effective at improving β -catenin levels (p.13) and cognitive capabilities, with a small significant increase in freezing during later training trials in the contextual fear conditioning task (p. 15-16), but no significant difference from vehicle-treated hets in the 24hr probe trial (Fig.8). We have expanded the text (p.15-16, 20-21).

Minor Concerns:

*1. Given the extensive focus on the hippocampus within this article, a rationale for not conducting behavioral tests relating to spatial and learning memory but via fear conditioning is warranted. Fear conditioning is a behavioral assay often employed to probe animal learning and memory particularly in the context of fear. Given the role of the amygdala in fear-related processes, it is pertinent to consider whether similar *Cttnb1* downregulation is observed in the amygdala of β -cat \pm mice.*

We have used contextual fear conditioning and 24 hr probe trials as these tasks assess associative learning and memory acquisition based on spatial cues (the context) and emotional experience and involve hippocampal-

amygdala circuits in the learning (encoding) and memory (retrieval) (Phillips and LeDoux, 1992; Xu et al., 2016; Kim and Cho, 2020). We have added a statement to this effect in the text (p.7). Although we have not directly measured β -catenin protein levels in the amygdala of the *Ctnnb1* het mouse, we expect that the germline heterozygosity affects all tissues.

2. The coognitive and motor impairments evident in β -cat^{+/-} mice underscore the relevance of assessing forelimb grip strength and rotarod performance. These evaluations could provide insights into potential enhancements facilitated by the dual GSK3 α and β inhibitor.

As suggested, we include new data that show significant improvements in forelimb grip strength and rotarod performance in GSK3 α,β dual paralog inhibitor treated *Ctnnb1* het mice, relative to vehicle treated hets (Fig.8 new panels B,C).

3. Notably absent from the study is data pertaining to the neuronal excitability of β -cat^{+/-} post-treatment with the GSK3 inhibitor.

As recommended, we include new data showing that the GSK3 α,β inhibitor improved neuronal excitability of the *Ctnnb1* hets, based on significant increases in spike frequency at higher current injections, rheobase and action potential waveform after hyperpolarization, relative to vehicle-treated hets (new Fig. 7).

Furthermore data characterizing molecular changes and behavioral outcomes subsequent to GSK3 inhibitor administration in β -cat^{+/-} mice are lacking.

The original manuscript showed that the GSK3 α,β inhibitor significantly normalized the molecular changes, including β -catenin, Na/K ATPase $\alpha 2$ and $\alpha 3$ levels (original Fig.6), and cognitive function, based on contextual fear conditioning and 24 hr probe trials (original Fig.7). In response to the review, we have added new data that further shows significant improvements in molecular, functional, and motor phenotypes, including increases in β -catenin association with N-cadherin synaptic adhesion protein, protein expression levels of Wnt targets DKK1 and EN2 (Fig. 6 new panels C,D), neuronal excitability (as indicated above; new Fig. 7), muscle grip strength, and rotarod performance (Fig. 8 new panels B,C).

*4. Cytosol and nucleus *Ctnnb1* is unregulated but suppressed synapse *Ctnnb1* in *Shank3* mice. But not as simply stated as down-regulated *Ctnnb1* in *Shank3* mice.*

Thank you for this correction. We have added additional references (Introduction, p.4) that report diverse effects of different *Shank3* mutations on levels of synaptic membrane β -catenin, nuclear β -catenin, and β -catenin-mediated Wnt target gene expression (Qin et al., 2018; Nia et al., 2020; Ioannidis et al., 2023).

Referee #2 (Comments on Novelty/Model System for Author):

The statistical tests are fine overall but there are some outliers restricted to the experimental animals that may drive the result. Some controls are missing. The mouse models themselves are not novel (have been previously generated) but the examination of them is. Medical impact is difficult to assess as only a single dose of inhibitor was given, no longer term effects measured.

We thank this reviewer for noting the novelty of our study of the *Ctnnb1* germline heterozygous mouse as a preclinical *in vivo* model of the human disorder CTNNB1 syndrome.

Previous studies of *Ctnnb1* heterozygous mouse models largely focused on embryonic development, with limited phenotypic characterization at later ages. Our study identifies novel molecular and functional changes,

and a small molecule treatment that significantly normalizes the molecular, functional, cognitive and motor phenotypes.

There are no outliers in any of the data sets that could drive the results. All data is based on quantitative assessments in PRISM and include ROUT tests to identify outliers. None were detected. We include new plots of the performance values of each individual mouse across all trials in the cognitive task, demonstrating that there is inter-mouse variability, but no consistently low underperformer (potential outlier) (New extended data Fig. 1). There was also no difference in performance between male and female mice (New extended data Fig. 1). Compared to WT littermates, the *Ctnnb1* het mice showed no significant difference in locomotion (velocity, total distance traveled) (Fig.8E). No mice were sick, and none exhibited a poor physical appearance or reduced general activity, our veterinary staff provide daily health checks.

It is not specified what controls are considered missing, so difficult to address.

We agree that dose-response studies and chronic treatment are important next steps, and we plan them as future studies. As this is the first evidence of a potential efficacious treatment for a severe developmental disorder lacking any treatment options, we feel it is important to share this data with the field as soon as possible. Further, there is precedence for the beneficial medical impact of long-term GSK3 α,β inhibition at appropriate doses stemming from individuals with psychiatric disorders being treated for decades with lithium salts, which include GSK3 α,β inhibition as one of its targets.

Referee #2 (Remarks for Author):

Here, the authors have generated a heterozygous mouse that lacks one copy of beta-catenin. The rationale was to mimic a human disorder observed relatively recently where inactivation of the human cognate occurs and that is associated with significant behavioural deficits. The haploinsufficient animals demonstrate certain behavioural changes compared to littermates that may relate to the patients phenotype and the authors show that these can be rescued by administration of a pan-isoform GSK3 inhibitor. This may have future implication for potential therapy.

1. It is perhaps surprising that there is a similar deficit in beta-catenin protein as RNA (which is expected) as shown in figure 1c. Typically, the majority of beta-catenin in cells is associated with the cadherins and excess is rapidly degraded via ubiquitin-mediated proteolysis via the destruction complex. The authors should estimate not only total beta-catenin but soluble (non-cadherin-associated beta-catenin) in addition to levels of N-cadherin and N-cadherin associated beta-catenin as they have done (in figure 3B).

As noted by the reviewer, b-catenin is present in multiple intracellular pools- cadherin associated adhesion complex, cytosolic, and nuclear pools. The *Ctnnb1* heterozygous mice display decreases in total b-catenin, b-catenin associated with N-cadherin, and expression levels of Wnt targets (Figs. 1,3). We find similar decreases in total b-catenin mRNA and protein levels in the mouse brain. Similar to this finding, previous reports show that b-catenin protein levels closely mirror mRNA levels in mouse embryonic stem cells (Rudloff and Kemler, 2012).

Ideally, this should be performed in the brain and at least one other organ (e.g. E-cadherin), given the whole body knockout. In this respect, the emphasis is almost exclusively on the brain, when there may be inter-organ aspects.

We agree that there may be inter-organ aspects as this is a germline haploinsufficiency. We have focused on changes in the brain because intellectual disabilities are a core phenotype in individuals with CTNNB1 syndrome. Fig.2 panels B,C shows motor deficits, reduced muscle grip strength and rotarod performance of Ctnnb1 heterozygous mice, relative to wildtype littermates, resembling motor deficits of poor coordination and muscle weakness in individuals with CTNNB1 syndrome.

2. The fear conditioning test should be supported by other short term memory tests such as Y maze. In general, the authors should perform an array of behavioural tests to assess the cognitive and other functions. As it stands, the data is rather minimal with respect to comparison with the human disorder.

In Fig 2, we show that, in comparison with wildtype littermates, Ctnnb1 heterozygous mice exhibit significantly reduced cognitive and motor capabilities, resembling the key features of human CTNNB1 syndrome. Our behavior tests include contextual fear conditioning (associative learning of a spatial cue, the context, with an emotional sensory experience, the foot shock), 24 hr probe trial (memory acquisition), rotarod (motor learning and coordination) and muscle grip strength.

3. As alluded to above, the Wnt pathway is well known for robust compensation. Changes in levels of components tend not to have large effects on output due to hysteresis feedback where several Wnt targets act to shut down signaling - e.g Axin 2 expression and FrzB hence limits to throughput are adapted. Plakoglobin has also been shown to be induced when beta-catenin is reduced. The authors have measured DKK1 and En2 but the signalling picture would be clearer with measurement of Axin2 and FrzB.

We agree that assessing changes in Wnt pathway feedback are informative. We have focused on the secreted Wnt antagonist DKK1 and on EN2, two canonical Wnt targets implicated in cognitive function. We show decreases in both DKK1 and EN2 protein expression levels in the Ctnnb1 heterozygous mouse hippocampus, relative to wildtype littermates (Fig.3C).

As suggested, we attempted to measure Axin2 and FrzB levels. Unfortunately, we were not able to detect a reproducible signal at the protein or mRNA levels by immunoblotting for Axin2 or RT-qPCR with validated primers for Axin2 and FrzB mRNAs in either the Ctnnb1 heterozygote or wildtype littermate hippocampus. According to the Allen Mouse Brain ISH Atlas, Axin2 and FrzB mRNA levels are very low in the mouse brain at the young adult ages that we are studying.

4. The GSK3 inhibitor data in figure 6 also raise some questions regarding levels of beta-catenin. They are observed to rise upon drug treatment but it is not shown whether this is the soluble and/or cadherin-associated beta-catenin. Given the majority of beta-catenin, as noted above, is cadherin-associated, inhibition of the kinase would not be expected to have a large affect on total beta-catenin. As in point 1, it is important to measure both populations (cadherin-associated and soluble).

As requested, we have added new data showing the GSK3 dual inhibitor leads to increases in total β -catenin, β -catenin associated with N-cadherin, and both DKK1 and EN2 protein expression levels in the Ctnnb1 heterozygotes such that the levels do not differ significantly from those of wildtype littermates (Fig. 6, new panels C, D). Precedence shows that inhibition of GSK3 α,β with BRD0320 increases total β -catenin protein and

Wnt signal transduction (β -catenin–TCF/LEF luciferase reporter assay) in human leukemia cell lines (Wagner et al., 2018).

5. *As mentioned in point 3, the effects of the inhibitors should include levels of Axin2 and/or FrzB to directly measure output of the Wnt pathway. This would also help address the important question of how much inhibition is required to compensate for the gene loss of beta-catenin given the authors are interested in possible therapeutic strategies. Given the isoform selective inhibitors individually do not reverse the impact of loss of the gene, this may suggest a high degree of inhibition is needed. If so, this may be problematic due to many other targets of the kinases.*

We agree that dose-response studies will be valuable for designing future therapeutic strategies and plan them in future studies of preclinical mouse and patient derived cell models. Support for the feasibility of long-term GSK3 α,β inhibition at appropriate doses stems from individuals with psychiatric disorders being treated for decades with lithium salts, which inhibit GSK3 α,β as one of its targets. Please also see reply above to Rev. 1, comment #4.

6. *Na/K ATPase alpha1 protein should be included in figure 6.*

As suggested, we include new data showing no significant difference in Na/K ATPase α 1 levels between GSK3 dual inhibitor-treated Ctnnb1 hets, vehicle-treated hets and vehicle-treated wildtype littermates (Fig. 6B)

Minor points:

A. *Both sexes were used in the behavioural experiments but were sufficient animals assessed to determine whether there were statistically significant differences based on sex?*

Both sexes were used in approximately equal numbers in all experiments- molecular, functional and behavioral assays of the het and WT littermate mice and no differences were detected based on sex, as now stated in the text (p.7) and shown in new data plots of the individual mouse performance across all trials and probe test in the contextual fear conditioning task (Extended View Fig.1). This finding is consistent with the lack of gender differences in the phenotypes of individuals with CTNNB1 syndrome.

B. *The distribution of individual animal results in figure 2A is unusual (note the +/- animals include several with very low freeze rates. Were these animals sick (there are reports of other effects of gene reduction of beta-catenin)? Is the statistical difference driven by these animals?*

Please see above response to Referee 2 Comments on Novelty/ Model System and the Extended View Fig.1.

Referee #3 (Comments on Novelty/Model System for Author):

More information on the natural history of disease progression in this animal model is required for evaluating the efficacy of corrective treatment at pre-symptomatic or symptomatic stage.

We agree that assessing the phenotypes and efficacy of corrective treatment at different developmental stages is important. We focused predominantly on cognitive deficits as this is a core phenotype in individuals with CTNNB1 syndrome. Our study used 6-10 wk old mice (late adolescence, young adult) as this developmental stage is the widely recommended standard for reliable quantitative assessment of complex cognitive tasks (contextual fear conditioning and probe trials) to assess learning and memory acquisition in mice. We show that the Ctnnb1 het mice at 6-10 weeks of age are symptomatic in that they display significant cognitive and motor deficits, as well as molecular and functional changes in the brain. This is demonstrated by comparison to littermate controls in every experiment, comparing the phenotypes of heterozygotes to WT littermates, and phenotypes of drug-treated heterozygotes to vehicle-treated heterozygote and vehicle-treated WT littermates.

As for disease progression in individuals with CTNNB1 syndrome, they display global developmental delays starting in the first months of life, not achieving developmental milestones (motor and language); the core cognitive and motor deficits are lifelong, and typically not progressive. The efficacy of the GSK3 α,β dual inhibitor to significantly improve the cognitive and motor deficits in the Ctnnb1 het mouse, when symptoms are apparent, is extremely encouraging and important to the families of individuals with this severe disorder.

Referee #3 (Remarks for Author):

On the premise that CTNNB1 syndrome is commonly associated with variants of CTNNB1 resulting in haploinsufficiency, the disease modelling study was undertaken in a heterozygous loss-of-function model. The findings point to the disruption in Na/K ATPase and synaptic adhesion are potential pathophysiological factors that underpin the altered electrochemical activity of cortical and hippocampal neurons in CTNNB1-related disease manifesting neurological dysfunction. By normalizing the level of CTNNB1 in the mutant animal and thereby restoring Na/K ATPase activity, through inhibiting glycogen synthase kinase activity, the molecular and neurological phenotypes can be reverted.

Points for clarification

Behavioral assessment:

- *For the contextual fear conditioning test and the rotarod test, clarify which datasets that were analyzed by multiple t-test, versus those by ANOVA, and then explain which of these two types of result of statistical significance is relevant to the interpretation of the outcome of the behavioral test.*

We apologize for the confusion likely caused by our reporting the ANOVA values in the text, and the multiple t-test values in the Fig. legends. We now report them both in the Fig. legends, in line with the journal policy. For all experiments, when either more than two groups are involved or when the same mice are being evaluated over separate trials (contextual fear conditioning, rotarod), we apply an ANOVA (either one-way ANOVA for a single comparison between more than two groups, or a repeated measure ANOVA to evaluate multiple groups of mice over repeated trials). When the ANOVA indicates a significant change, we apply multiple comparison t-tests to evaluate the significance of the group variability. These statistical analyses showed significant decreases in the cognitive, motor learning and coordination capabilities of the Ctnnb1 het mice, relative to WT littermates.

How was the percent freezing determined (Fig 2A, 7) - did some mutant mice not display any defect in contextual fear conditioning?

For each mouse, freezing scores were calculated by averaging freezing during minutes 2 and 3 of each trial, with 1 hour spacing between trials. Freezing behavior was measured using a digital camera connected to a computer with Actimetrics Freeze Frame software. The bout length was 1s and the threshold for freezing behavior was determined by an experimenter blind to experimental conditions, such as genotypes and drug- versus vehicle-treatments. We have added these details to the Methods (p.23).

We now include plots of the performance values of each individual heterozygote and wildtype mouse in all trials and probe test in the contextual fear conditioning assay, demonstrating variability, but no heterozygous mouse displayed normal performance levels across the assay (New extended data Fig. 1). There are no outliers in any of the data sets. All data is based on quantitative assessments in PRISM and include ROUT tests to identify outliers. None were detected.

- *For the forelimb grip-strength test, clarify the discrepancy in the reported p-value ($p=0.003$ average of 3 trials - page 6 paragraph 2) and the figure ($* p<0.001$) to ensure accuracy in the statistical analysis.*

Thank you for noting this mismatch, we have now corrected the p-value in the Fig.2 legend, it is $p=0.003$, not $p<0.001$.

- *For the open field test. clarify whether other parameters. such as total distance travelled or distance travelled in the center. were measured in the open field. These additional parameters are relevant in assessing the animals' exploratory and anxiety-related behaviors. which are of interest to establish CTNNB1-like phenotype, and their inclusion could strengthen the study's findings.*

We now state in the text (p. 7-8) that the open field test measured velocity of movement (total distance travelled over time), and show no significant difference between Ctnnb1 het mice and WT littermates (Fig. 2D). As recommended, we assessed anxiety using the elevated plus maze task. There was no significant difference between the het mice and WT littermates in the total number of entries or time spent in the open arm, or in the total distance travelled (velocity of movement), as shown in new data (Fig.8 D,E). Further, the GSK3 dual inhibitor did not alter the behavior of the het mice in this assay.

Functional property of hippocampal neuron:

- *It was indicated that the assays of excitability were performed on CA1 pyramidal neuron and dentate gyrus granule cells (DGGCs) (page 8 paragraph 2), but only the data of DGGCs were presented in Figure 4. What were the data ("similar functional changes" - figure legend and page 8 paragraph 2) of the CA1 pyramidal neurons omitted?*

We deleted the CA1 pyramidal neuron statement because our "n" was relatively small and we focused the efforts of our collaborator on generating the new data in Fig. 7 that shows the GSK3 dual inhibitor significantly improves the altered functional properties and excitability of the DGGCs in the Ctnnb1 hets, relative to vehicle-treated hets.

Levels of the three Na/K ATPase catalytic units were shown for the cortex and hippocampus, but not the "frontal cortex" (Fig. 5). Which part/s of the "cortex" that was/were outside the "frontal cortex" were studied? Why were the "similar" frontal cortex data not shown? "Data not shown" is not acceptable if the data were an integral part of results, especially that the impact of GSK3 inhibition was evaluated on "undefined" parts (or neuronal types) in the frontal cortex and hippocampus (Fig. 6).

We apologize for the confusion caused by not clarifying that we always used the frontal cortex but referred to it as "cortex" in parts of the manuscript. We have corrected this to clearly state frontal cortex.

GSK3 Rescue:

• *It was noted that "for all experiments 2-3-month-old mice of both sexes were used" (materials and methods), i.e., 8-12-week-old mice were used. Instead, the rescue experiments were performed on 6-weeks old mice (figure 6 legend). How was this timing chosen? No information of the natural history of cognitive and motor defect in this model was provided to indicate whether the mutant mice were symptomatic or pre-symptomatic for behavioral defect and neuronal dysfunction and if the phenotype is fully penetrant. This is a critical point to consider in the light of the assertion (Discussion) of significant corrective outcome with treatment "at young adult ages when the phenotypes are clearly displayed" (page 18) and that "These findings suggest a relatively wide age window for therapeutic intervention by showing that treatment initiated after the onset of symptoms still provides significant corrective outcomes" (page 15).*

We now clarify that all experiments in this study used mice from 6-10 weeks of age. Mice in this age range were treated with the GSK3 α,β inhibitor for 5 days prior to tests for corrective outcomes. We have revised Fig.6A and the text to clarify this. We used 6-10 week-old mice (late adolescence, young adult) as this developmental stage is the widely recommended standard for reliable quantitative assessment of complex cognitive tasks in mice.

The Ctnnb1 het mice at 6-10 weeks of age were symptomatic in that they displayed significant cognitive and motor deficits, as well as molecular and functional changes. This is demonstrated by comparison to littermate controls in every experiment, comparing the phenotypes of heterozygotes to WT littermates, and phenotypes of drug-treated heterozygotes to vehicle-treated heterozygote and WT littermates.

Our data show that the GSK3 α,β dual inhibitor significantly improved the phenotypes of the Ctnnb1 het mice at this age, when symptoms are apparent. Further, we have revised the statement (p.5,17) to now state that the therapeutic treatment was effective when administered in symptomatic mice.

Specific points-

Indicate the age of the animals analyzed for behavioral phenotypes, and those from which tissues were sampled for biochemistry studies (materials and methods).

For all experiments, mice were analyzed at 6-10 weeks-of-age with Ctnnb1 hets compared to WT littermates, and GSK3 inhibitor treated hets compared to vehicle treated het and WT littermates.

Text:

• *The labels of the panels (in lower case) did not match those in the legend (in capitals)*

We have corrected the panel labels to all be upper case.

• *Clarify whether the correct p-value is $p < 0.01$ or $p < 0.001$ (page 6, line 8) for over the training trial results to ensure accuracy in the statistical analysis.*

The correct p values are listed for each trial in Fig. 2 legend.

- *It would be appropriate to (i) revise the section heading to: "... improves the cognitive function of β -cat^{+/-} mice" (page 12) and (ii) present the statistical results in the figure 7 legend or in tabulated form in the Fig 7, and not in the text (page 13 paragraph 2).*

We have revised the heading to state "significantly improves the cognitive and motor capabilities of the β -cat het mice". We have improved the flow by moving the statistical results to the figure legends.

- *The sentence following "In summary" (page 19) is a re-iteration of the introduction and does not belong to the summary.*

We have deleted this sentence.

Figures:

- *Fig. 1 and 2 Address the inconsistency in the reported p-value on page 21, line 17, and page 6, regarding the statistical significance The correct and consistent p-value should be provided to avoid any potential misinterpretation of the study results.*

We have corrected the typo in the number of asterisks for the p value on p.21; all correct p values are stated in the figure legends in the revised text.

- *Fig. 2: It is essential to explain why there is a discrepancy in the number of mice shown in the graphs of Figure 2 if the tests were performed on the same animals. This inconsistency might impact the reliability of the results, and further clarification is needed. Fig 2D: The inconsistency in the reported sample size should be addressed. If the control group indeed consists of 9 values, the correct sample size should be stated accurately as "n=9" on page 6 last line to avoid confusion.*

We apologize for the confusion, the same mice were used for the tasks in Fig.2, but the total n varies slightly between the tasks based on the time we allotted for our use of the equipment in the Behavior Core. The n=12 per genotype for contextual fear conditioning and 24 hr probe test (panel A), n=9 WT littermates and n=10 *Ctnnb1* hets for the forelimb grip strength and open field (panels B,D) and n=10 per genotype for the rotarod (panel C).

- *Fig. 3: Clarify (i) which of the double protein bands of N-cadherin and LEF1 were subjected to quantification (Fig. 3A), (ii) the apparent inconsistency of the IP data (Fig. 3B) and the quantified results, and (iii) the discrepancy of the quantified data with the blot data of DKK1 protein (Fig. 3C).*

Both double bands were quantified to measure the total levels of α -N-catenin and LEF1 protein (Fig. 3A). We repeated the N-cadherin and β -catenin co-immunoprecipitation analysis with new tissue samples and provide an improved representative immunoblot image (panel 3B). The discrepancy between the blot and histogram data in panel 3C is not clear to us.

- *Fig. 4: Explain the discrepancy between the stated sample size on page 8 and the number of data points shown in Figure 4. If the graphs display more than eight (8) values per group (or n=4 independent samples with 2 replicates each), the authors should clarify the reason for the inclusion of*

more data points in the figure to ensure the accuracy of the data presentation. For Fig. 4B, and 4E: Provide the colour legend of the two tracings.

We apologize for the error in the original text stating n=4 mice per genotype, whereas the correct values are n = 5 controls and 6 β -cat hets, 2 cells each mouse, as shown in the Fig.4 data and now correctly stated in the revised Fig.4 legend. Further, we have added the color legend to Fig.4 B, E panels.

Fig. 2A, 6 and 7: Indicate in the figures which pairs of datasets were assessed (e.g., Supplementary figure 1A) to reach the respective statistical results

We have added lines to demarcate which datasets are being compared in all figures.

10th Jun 2024

Dear Dr. Jacob,

Thank you again for submitting your revised work to EMBO Molecular Medicine. We have now heard back from the original three reviewers who evaluated your study.

As you will see below, the reviewers are supportive on the novelty and potential therapeutic benefits of the GSK3 dual inhibitor, but still have some remaining concerns on its efficacy and potential medical impact. Although Reviewer 3 has pointed out that additional longer timepoints after treatment would be necessary, editorially we find the single 1 hour post-treatment sufficient. However, we would agree that the comparison to vehicle-treated WT mice should be included to determine the functional consequences of the therapy in Figure 7. Additional examination in other tissues would be helpful to further validate the model and strengthen the manuscript. In line with Reviewer 2's comments that the mechanism of rescue is still unclear, it will be necessary to tone down the conclusions and discuss the limitations.

Addressing the remaining reviewers' concerns in full in a point-by-point response will be necessary for further considering the manuscript in our journal. We are expecting your re-revised manuscript within three months.

We remind you that we have the following formatting requirements:

4) A .docx formatted letter INCLUDING the reviewers' reports and your detailed point-by-point responses to their comments. As part of the EMBO Press transparent editorial process, the point-by-point response is part of the Review Process File (RPF), which will be published alongside your paper.

5) A complete author checklist, which you can download from our author guidelines (<https://www.embopress.org/page/journal/17574684/authorguide#submissionofrevisions>). Please insert information in the checklist that is also reflected in the manuscript. The completed author checklist will also be part of the RPF.

6) Please note that all corresponding authors are required to supply an ORCID ID for their name upon submission of a revised manuscript.

7) It is mandatory to include a 'Data Availability' section after the Materials and Methods. Before submitting your revision, primary datasets produced in this study need to be deposited in an appropriate public database, and the accession numbers and database listed under 'Data Availability'. Please remember to provide a reviewer password if the datasets are not yet public (see <https://www.embopress.org/page/journal/17574684/authorguide#dataavailability>).

This study includes no data deposited in external repositories.

8) For data quantification: please specify the name of the statistical test used to generate error bars and P values, the number (n) of independent experiments (specify technical or biological replicates) underlying each data point and the test used to calculate p-values in each figure legend. The figure legends should contain a basic description of n, P and the test applied. Graphs must include a description of the bars and the error bars (s.d., s.e.m.). Please provide exact p values.

- the medical issue you are addressing,

- the results obtained and

- their clinical impact.

13) Author contributions: CRedit has replaced the traditional author contributions section because it offers a systematic machine readable author contributions format that allows for more effective research assessment. Please remove the Authors Contributions from the manuscript and use the free text boxes beneath each contributing author's name in our system to add specific details on the author's contribution. More information is available in our guide to authors.

Please also suggest a visual abstract to illustrate your article as a jpeg file 550 px wide x 300-600 px high.

Share synopsis text and image, as well as eTOC:

Please note that these would be the final versions and changes during proofing are usually not allowed

16) As part of the EMBO Publications transparent editorial process initiative (see our policy here:

https://www.embopress.org/transparent-process#Review_Process), EMBO Molecular Medicine will publish online a Peer Review File (PRF) to accompany accepted manuscripts.

In the event of acceptance, this file will be published in conjunction with your paper and will include the anonymous referee reports, your point-by-point response and all pertinent correspondence relating to the manuscript. Let us know whether you agree with the publication of the PRF and as here, if you want to remove or not any figures from it prior to publication.

I look forward to receiving your revised manuscript.

Yours sincerely,

Poonam Bheda

Poonam Bheda, PhD
Scientific Editor
EMBO Molecular Medicine

**** Reviewer's comments ****

Referee #1 (Comments on Novelty/Model System for Author):

NS

Referee #1 (Remarks for Author):

New data have been added for further clarification, especially the potential mechanisms of treatment effects of dual inhibition of both GSK3 α and GSK3 β . Additional explanation and discussion have been added in the main text. All my concerns have been addressed and responded in detail.

Referee #2 (Comments on Novelty/Model System for Author):

The model chosen is appropriate and reflects the human disorder of haploinsufficiency. Medical impact is hard to judge given the lack of information on the longer-term nature of the require kinase inhibition and the other impacts this will likely have (on pathways other than Wnt and cadherin). Novelty is medium to high in that the model itself isn't new, nor is the kinase inhibitor, but the combination yields new insights.

Referee #2 (Remarks for Author):

The authors have addressed a number of the issues raised and clarified several questions. The lack of detection of Axin2 and FrzB in the brain is what it is but the point of investigating those levels was to provide more insight into the impact on Wnt signaling and the effect of the kinase inhibitor. As it stands, it is somewhat open as to whether the rescue effect is due to Wnt or cadherin. It is also critical to determine the tolerance of the animals for the inhibitor given this is proposed as a possible therapeutic against a quite complicated series of kinase targets outside of beta-catenin signaling. Still, given the condition has no current therapeutic option, this will have to be explored in further preclinical studies beyond this report.

Referee #4 (Comments on Novelty/Model System for Author):

The murine animal model presented in the article appears suitable for representing phenotypes found in CTNNB1 Syndrome patients. However, there are several critical areas that need further examination to fully validate this model. Specifically, the levels of β -catenin were only assessed in two regions of the brain. To comprehensively determine the suitability of this model, it is essential to assess β -catenin levels in other tissues, such as muscle.

The medical impact is not clear at this stage. The efficacy of the GSK3 dual inhibitor, the authors present protein expression levels that are comparable to control wildtype mice and show improvement compared to untreated mice (Figure 6). However, when assessing functional properties and excitability (Figure 7), the comparison against wildtype control mice is missing. This omission makes it difficult to determine whether the observed phenotype was corrected. Including a wildtype control group in these comparisons is crucial for a thorough evaluation. Furthermore, the study lacks data beyond one hour post-treatment. Assessments at later time points are necessary to evaluate the efficacy and potency of the therapeutic candidate.

Despite these areas for improvement, the study is novel in its identification of molecular phenotypes that serve as therapeutic targets for CTNNB1 Syndrome and the use of a GSK3 dual inhibitor to correct these phenotypes. Thus, this research offers valuable insights into potential treatment avenues for this condition.

Referee #4 (Remarks for Author):

The study is novel in identifying molecular phenotypes as therapeutic targets for CTNNB1 Syndrome and using a GSK3 dual inhibitor, offering valuable insights into potential treatments for CTNNB1 Syndrome. The murine model in the article appears suitable for representing CTNNB1 Syndrome phenotypes but requires further validation. Specifically, β -catenin levels were only assessed in two brain regions, and assessment in other tissues like muscle is necessary.

The medical impact remains unclear as the efficacy data lacks long-term post-treatment assessments and comparisons against wildtype control mice for functional properties in Figure 7.

The authors should provide more detailed information on the statistical methods employed, including the randomization of mice, power analysis, assessment of normality, and the specific post-hoc tests used. Additionally, details about the quantification of the immunoblots should be included; if image-based quantification was used, it should be clarified that this constitutes semi-quantification.

In terms of data presentation, there are sections where results are discussed without accompanying figures, specifically in the last paragraph of page 12, the first paragraph of page 13, and on page 16, lines 1-4. Including figures in these sections would enhance clarity and support the findings more robustly. The co-immunoprecipitation shown in Figure 3B needs to be redesigned to enhance the clarity and understandability of the results and the quantification should be accompanied by individual values and error bars in the control group.

Consider adding figures that detail the protocols for mouse model generation and experimental procedures as supplementary material. It would also be beneficial to include precedents for the functional changes observed or to add these figures as supplementary material.

There are some misspellings in the text that need correction: "by" should start with a capital letter on page 17, line 22. Additionally, in the legend of extended figure 2A, "(Vehcile)" should be corrected to "(Vehicle)". There is inconsistency in the use of the abbreviation for β -catenin (e.g., "b-cat" or "bcat") on page 9, which should be standardized throughout the manuscript. Moreover, there are some unrecognized symbols in the text legends for figures 3 (line 4) and 6 (line 2) in the PDF document, which need correction.

Finally, the discussion section would benefit from a more thorough exploration of the impact of these results, their potential for translation to clinical settings, and the safety implications of chronic exposure to the therapeutic candidate. While the study makes significant contributions, addressing these points would strengthen the conclusions and provide a more comprehensive understanding of the model and the therapeutic candidate's potential.

EMM-2023-18356-V3 Decision Letter

10th Jun 2024

Dear Dr. Bheda,

Thank you for this opportunity to respond to the remaining concerns of Reviewer 2 and 3. We now include the additional data requested by Reviewer 3 and have added text as suggested by Reviewer 2. Below, we detail our response to each comment.

Poonam Bheda, PhD
Scientific Editor
EMBO Molecular Medicine

"As you will see below, the reviewers are supportive on the novelty and potential therapeutic benefits of the GSK3 dual inhibitor, but still have some remaining concerns on its efficacy and potential medical impact. Although Reviewer 3 has pointed out that additional longer timepoints after treatment would be necessary, editorially we find the single 1 hour post-treatment sufficient."

We appreciate this editorial support. Please note that the cognitive phenotype was assessed up to 24 hours post-treatment (Fig. 8A). We start assessing the phenotypes at 1 hour post-treatment. The contextual fear conditioning assay for associative learning requires training trials over hours and includes a 24 hour probe trial for memory acquisition, that showed the cognitive performance of the GSK3 α,β inhibitor treated heterozygote mice was not significantly different from that of vehicle treated wildtype littermates.

"However, we would agree that the comparison to vehicle-treated WT mice should be included to determine the functional consequences of the therapy in Figure 7."

We now include this comparison, new Extended View Figure 4. The improved excitability of the GSK3 α,β inhibitor treated heterozygote mice is not significantly different from the excitability of wildtype littermates, based on comparisons of spike frequency (max firing rate at increasing current injections) and relative levels of rheobase currents necessary to elicit an action potential.

"Additional examination in other tissues would be helpful to further validate the model and strengthen the manuscript."

We now include new data Extended View Figure 1 that shows approximately 50% decreases in β -catenin protein levels in skeletal muscle (diaphragm) of Ctnnb1 heterozygous mice, relative to wildtype littermates, and significant normalization of the β -catenin levels by treatment with the GSK3 dual inhibitor.

"In line with Reviewer 2's comments that the mechanism of rescue is still unclear, it will be necessary to tone down the conclusions and discuss the limitations."

We have added new text, p.21(highlighted) that discusses the mechanism of rescue and the limitations.

***** Reviewer's comments *****

Referee #1 (Comments on Novelty/Model System for Author):

NS

Referee #1 (Remarks for Author):

New data have been added for further clarification, especially the potential mechanisms of treatment effects of dual inhibition of both GSK3 α and GSK3 β . Additional explanation and discussion have been added in the main text. All my concerns have been addressed and responded in detail.

We thank you for your previous comments that helped to improve the manuscript and for your appreciation of the new data and additional explanations.

Referee #2 (Comments on Novelty/Model System for Author):

"The model chosen is appropriate and reflects the human disorder of haploinsufficiency. Medical impact is hard to judge given the lack of information on the longer-term nature of the require kinase inhibition and the other impacts this will likely have (on pathways other than Wnt and cadherin). Novelty is medium to high in that the model itself isn't new, nor is the kinase inhibitor, but the combination yields new insights."

Thank you for your helpful comments and for stating the novelty is medium to high. Our preclinical in vivo mouse data provide the first evidence for a treatment that significantly corrects cognitive and motor phenotypes relevant to this severe neurodevelopmental disorder that lacks current treatment options.

Referee #2 (Remarks for Author):

"The authors have addressed a number of the issues raised and clarified several questions. The lack of detection of Axin2 and FrzB in the brain is what it is but the point of investigating those levels was to provide more insight into the impact on Wnt signaling and the effect of the kinase inhibitor. As it stands, it is somewhat open as to whether the rescue effect is due to Wnt or cadherin."

Normalizing β -catenin protein levels and associated functions in both the cadherin based synaptic adhesion complex and the canonical Wnt signal transduction pathway is the most likely mechanism of rescue, based on significant corrections by the GSK3 dual inhibitor treatment (Fig. 6C and D). We cannot delineate their relative contributions to the phenotype corrections or rule out a potential contribution of β -catenin independent effects of the GSK3 dual inhibitor. We have now added this statement to the text, p. 21.

"It is also critical to determine the tolerance of the animals for the inhibitor given this is proposed as a possible therapeutic against a quite complicated series of kinase targets outside of beta-catenin signaling. Still, given the condition has no current therapeutic option, this will have to be explored in further preclinical studies beyond this report."

We agree and state the need for future studies "to evaluate the safety of chronic treatment with the highly selective GSK3 α,β inhibitor. As a first step, we show no significant molecular or cognitive changes in wildtype mice with the same dual inhibitor treatment paradigm that provided corrective outcomes in the β -catenin het mice" text p.22, Extended View Fig.5.

Precedence shows other GSK3 α,β inhibitors have advanced to Phase 2 trials in adults and Phase 2/3 trials in children and adolescents for other conditions (Rizzieri et al., 2016; Horrigan et al., 2020; Clinical trial NCT05004129). Further, support for the feasibility of long-term GSK3 inhibition at appropriate doses stems from individuals with psychiatric disorders being treated for decades with lithium salts, which include GSK3 inhibition as one of its targets. Based on its high selectivity for GSK3 against the human kinome, blood brain barrier permeability and physicochemical properties, we believe that our small molecule GSK3 inhibitor is best for developing an efficacious and safe treatment for CTNNB1 Syndrome patients. This statement is added to p. 17-18.

Referee #3 (Comments on Novelty/Model System for Author):

"The murine animal model presented in the article appears suitable for representing phenotypes found in CTNNB1 Syndrome patients. However, there are several critical areas that need further examination to fully validate this model. Specifically, the levels of β -catenin were only assessed in two regions of the brain. To comprehensively determine the suitability of this model, it is essential to assess β -catenin levels in other tissues, such as muscle."

As suggested, we now include Ctnnb1 heterozygous mouse skeletal muscle (diaphragm) data that shows approximately 50% decreases in β -catenin protein levels and significant normalization of the β -catenin levels by treatment with the GSK3 dual inhibitor, new Extended View Figure 1.

"The medical impact is not clear at this stage. The efficacy of the GSK3 dual inhibitor, the authors present protein expression levels that are comparable to control wildtype mice and show improvement compared to untreated mice (Figure 6). However, when assessing functional properties and excitability (Figure 7), the comparison against wildtype control mice is missing. This omission makes it difficult to determine whether the observed phenotype was corrected. Including a wildtype control group in these comparisons is crucial for a thorough evaluation."

As recommended, we now compare the improved excitability of the GSK3 α,β inhibitor treated heterozygote mice to the excitability of wildtype littermates. There is no significant difference based on comparisons of spike frequency (max firing rate) at increasing current injections and relative levels of rheobase currents necessary to elicit an action potential, new Extended View Figure 4.

As further evidence of the potential therapeutic efficacy of the GSK3 dual inhibitor, we show significantly normalized cognitive and motor phenotypes in the inhibitor treated heterozygote mice, relative to vehicle-treated wildtype littermates (Fig.8).

"Furthermore, the study lacks data beyond one hour post-treatment. Assessments at later time points are necessary to evaluate the efficacy and potency of the therapeutic candidate."

The cognitive phenotype was assessed up to 24 hours post-treatment (Fig. 8A). We start assessing the phenotypes at 1 hour post-treatment. The contextual fear conditioning assay for associative learning requires training trials over hours and includes a 24 hour probe trial for memory acquisition, that showed the cognitive performance of the GSK3 α,β inhibitor treated heterozygote mice was not significantly different from that of vehicle treated WT littermates.

"Despite these areas for improvement, the study is novel in its identification of molecular phenotypes that serve as therapeutic targets for CTNNB1 Syndrome and the use of a GSK3 dual inhibitor to correct these phenotypes. Thus, this research offers valuable insights into potential treatment avenues for this condition".

Thank you for your suggestions that improve the data and for noting the novelty and importance of our findings.

Referee #3 (Remarks for Author):

"The study is novel in identifying molecular phenotypes as therapeutic targets for CTNNB1 Syndrome and using a GSK3 dual inhibitor, offering valuable insights into potential treatments for CTNNB1 Syndrome. The murine model in the article appears suitable for representing CTNNB1 Syndrome phenotypes but requires further validation. Specifically, β -

catenin levels were only assessed in two brain regions, and assessment in other tissues like muscle is necessary."

*As recommended, we now include skeletal muscle (diaphragm) data that shows approximately 50% decreases in β -catenin protein levels in *Ctnnb1* heterozygous mice, relative to wildtype littermates, and significant normalization of the β -catenin levels by treatment with the GSK3 dual inhibitor, new Extended View Figure 1.*

"The medical impact remains unclear as the efficacy data lacks long-term post-treatment assessments and comparisons against wildtype control mice for functional properties in Figure 7."

As detailed above, we now compare the improved excitability of the drug-treated het mice and excitability of wildtype littermates, new Extended View Figure 4.

"The authors should provide more detailed information on the statistical methods employed, including the randomization of mice, power analysis, assessment of normality, and the specific post-hoc tests used.

All figure legends include the specific statistical methods and post-hoc tests used for all data. We have added detail to the methods that address the randomization of mice, power analysis, and assessment of normality (p. 27).

Additionally, details about the quantification of the immunoblots should be included; if image-based quantification was used, it should be clarified that this constitutes semi-quantification.

We have expanded the details for quantification of the immunoblots (p. 26).

"In terms of data presentation, there are sections where results are discussed without accompanying figures, specifically in the last paragraph of page 12, the first paragraph of page 13, and on page 16, lines 1-4. Including figures in these sections would enhance clarity and support the findings more robustly."

As suggested, we now present the data in new Extended View Figures 3, 6.

"The co-immunoprecipitation shown in Figure 3B needs to be redesigned to enhance the clarity and understandability of the results and the quantification should be accompanied by individual values and error bars in the control group."

We detail the quantification in the Methods section (p. 26-27) to improve clarity.

"Consider adding figures that detail the protocols for mouse model generation and experimental procedures as supplementary material."

The protocol for generating the mouse model is shown in Fig. 1A and described in the text (p. 6, 23).

"It would also be beneficial to include precedents for the functional changes observed or to add these figures as supplementary material."

This is the first study, to our knowledge, of functional changes in Ctnnb1 heterozygote mice.

"There are some misspellings in the text that need correction: "by" should start with a capital letter on page 17, line 22. Additionally, in the legend of extended figure 2A, "(Vehcile)" should be corrected to "(Vehicle)". There is inconsistency in the use of the abbreviation for β -catenin (e.g., "b-cat" or "bcat") on page 9, which should be standardized throughout the manuscript. Moreover, there are some unrecognized symbols in the text legends for figures 3 (line 4) and 6 (line 2) in the PDF document, which need correction."

We have corrected these typos.

"Finally, the discussion section would benefit from a more thorough exploration of the impact of these results, their potential for translation to clinical settings, and the safety implications of chronic exposure to the therapeutic candidate. While the study makes significant contributions, addressing these points would strengthen the conclusions and provide a more comprehensive understanding of the model and the therapeutic candidate's potential."

We have expanded the discussion (p. 17-18), as suggested here, and by Reviewer 2.

5th Jul 2024

Dear Dr. Jacob,

Thank you for the submission of your revised manuscript to EMBO Molecular Medicine. We have now received the enclosed reports from the referees that were asked to re-assess it. As you will see the reviewers are now globally supportive and I am pleased to inform you that we will be able to accept your manuscript pending the following final amendments:

- 1) In the main manuscript file, please include keywords to max. 5.
- 2) As there are no data that are required to be uploaded to external repositories, please use the following sentence for the Data Availability statement: "This study includes no data deposited in external repositories."
- 3) Please rename "Conflict of Interest" to "Disclosure and competing interests statement". We updated our journal's competing interests policy in January 2022 and request authors to consider both actual and perceived competing interests. Please review the policy <https://www.embopress.org/competing-interests> and update your competing interests if necessary.
- 4) In the Methods, please take care of the following:
 - Please rename this section to "Methods"
- 5) All Materials and Methods need to be described in the main text using our 'Structured Methods' format, which is required for all research articles. According to this format, the Methods section includes a Reagents and Tools Table (listing key reagents, experimental models, software and relevant equipment and including their sources and relevant identifiers) followed by a Methods and Protocols section describing the methods using a step-by-step protocol format. The aim is to facilitate adoption of the methodologies across labs. More information on how to adhere to this format as well as a downloadable template (.docx) for the Reagents and Tools Table can be found in our author guidelines:
<https://www.embopress.org/page/journal/17574684/authorguide#structuredmethods>
An example of a Method paper with Structured Methods can be found here:
<https://www.embopress.org/doi/10.15252/msb.20178071>.
- 6) Please place individual sections of the manuscript in the following order: Title page - Abstract & Keywords - Introduction - Results - Discussion - Materials & Methods - Data Availability - Acknowledgements - Disclosure and Competing Interests Statement - The Paper Explained - For More Information - References - Figure Legends - Expanded View Figure Legends.
- 7) For the figures and figure legends, please take care of the following:
 - Please rename the callouts for the Expanded View figures according to our formatting - e.g. please use "Figure EV1" instead of "Extended View Fig.1". Please also update the name in the legend to "Expanded View Figure Legends".
 - Please note that we require exact p-values to be reported in either the figure or figure legend. Currently not all exact p-values are provided.
 - Please indicate the statistical test used for data analysis in the legends of figures 2d; 3a, supplementary figures 1a-b.
 - Please note that in figure 2b there is a mismatch between the annotated p values in the figure legend and the annotated p values in the figure file that should be corrected.
 - Please note that information related to n is missing in the legends of figures 1b-c; 2a-d; 3a-c; 4a, c-d; 5; 6a-b; 7, supplementary figures 1a-b.
 - Please note that the error bars are not defined in the legends of figures 1b-c; 2a-d; 3a-c; 4a, c-d; 5; 6a-b; 7, supplementary figures 1a-b.
- 8) Funding: Please ensure that all funding sources are entered into the manuscript submission system - Advancing CTNNB1 Cures and Treatments, Inc. (currently CTNNB1 Connect and Cure) (to MHJ) needs to be entered as a separate funder.
- 9) Synopsis:
 - Synopsis text: Please shorten the standfirst to a maximum of 300 characters, including spaces.
 - Please check your synopsis text and image before submission with your revised manuscript. Please be aware that in the proof stage minor corrections only are allowed (e.g., typos).
- 10) Source Data: Please ensure that all Source Data are provided. We uploaded the Source Data provided by email to your submission. However, when we previously requested Source Data for Figure 6C due to an image check, the source data provided seems to be for Figure 6D and not Figure 6C. Please double check that the correct Source Data are uploaded to your submission. In addition, Source Data for Figure 8 seems to be missing.
- 11) As part of the EMBO Publications transparent editorial process initiative (see our policy here: https://www.embopress.org/transparent-process#Review_Process), EMBO Molecular Medicine will publish online a Peer Review File (PRF) to accompany accepted manuscripts. This file will be published in conjunction with your paper and will include the anonymous referee reports, your point-by-point response and all pertinent correspondence relating to the manuscript. Let us know whether you agree with the publication of the PRF and as here, if you want to remove or not any figures from it prior to publication. Please note that the Authors checklist will be published at the end of the PRF.
- 12) Please provide a point-by-point letter INCLUDING my comments as well as the reviewer's reports and your detailed responses (as Word file).

I look forward to reading a new revised version of your manuscript as soon as possible.

Yours sincerely,

Poonam Bheda

Poonam Bheda, PhD
Scientific Editor
EMBO Molecular Medicine

***** Reviewer's comments *****

Referee #2 (Comments on Novelty/Model System for Author):

The mouse model is a reasonable mimic for the human deficiency in CTNNB1 syndrome as shown by the data in this manuscript. Medical impact is hard to judge and beyond this study.

Referee #2 (Remarks for Author):

The authors have included additional information and clarifications that address the remaining concerns.

Referee #4 (Comments on Novelty/Model System for Author):

The authors addressed the raised issues, adhered to recommendations, and clarified the questions. They demonstrated the downregulation of β -catenin in an additional tissue (muscle), thereby confirming the model's adequacy. They compared the excitability of heterozygous mice treated with the GSK3 dual inhibitor to that of wild-type littermates to assess phenotype correction, highlighting a potential therapeutic impact. Furthermore, they provided detailed information on the statistics and methods used, and included previously missing data, enhancing the technical quality of the study.

Referee #4 (Remarks for Author):

The authors addressed the issues raised, followed recommendations, and clarified the questions. They showed the downregulation of β -catenin in other tissues such as the muscle, demonstrating suitability of the model. They also compared the excitability of het mice treated with the GSK3 dual inhibitor against WT littermates to determine phenotype correction. Furthermore, they provided details about statistics and methods developed and included some data that was missing. There is still a typo in page 17, 7th line from the bottom "reduced. by", they should remove the full stop.

***** Reviewer's comments *****

Referee #2 (Comments on Novelty/Model System for Author):

The mouse model is a reasonable mimic for the human deficiency in CTNNB1 syndrome as shown by the data in this manuscript. Medical impact is hard to judge and beyond this study.

Referee #2 (Remarks for Author):

The authors have included additional information and clarifications that address the remaining concerns.

Thank you.

Referee #4 (Comments on Novelty/Model System for Author):

The authors addressed the raised issues, adhered to recommendations, and clarified the questions. They demonstrated the downregulation of β -catenin in an additional tissue (muscle), thereby confirming the model's adequacy. They compared the excitability of heterozygous mice treated with the GSK3 dual inhibitor to that of wild-type littermates to assess phenotype correction, highlighting a potential therapeutic impact. Furthermore, they provided detailed information on the statistics and methods used, and included previously missing data, enhancing the technical quality of the study.

Thank you.

Referee #4 (Remarks for Author):

The authors addressed the issues raised, followed recommendations, and clarified the questions. They showed the downregulation of β -catenin in other tissues such as the muscle, demonstrating suitability of the model. They also compared the excitability of het mice treated with the GSK3 dual inhibitor against WT littermates to determine phenotype correction. Furthermore, they provided details about statistics and methods developed and included some data that was missing. There is still a typo in page 17, 7th line from the bottom "reduced. by", they should remove the full stop.

We have corrected this typo.

17th Jul 2024

Dear Dr. Jacob,

We are pleased to inform you that your manuscript is accepted for publication and is now being sent to our publisher to be included in the next available issue of EMBO Molecular Medicine.

Yours sincerely,

Poonam Bheda, PhD
Scientific Editor
EMBO Molecular Medicine
